# Wasserstein Distortion: Unifying Fidelity and Realism

## Abstract

We introduce a distortion measure for images, Wasserstein distortion, that simultaneously generalizes pixel-level fidelity on the one hand and realism or perceptual quality on the other. We show how Wasserstein distortion reduces mathematically to a pure fidelity constraint or a pure realism constraint under different parameter choices. Pairs of images that are close under Wasserstein distortion illustrate its utility. In particular, we generate random textures that have high fidelity to a reference texture in one location of the image and smoothly transition to an independent realization of the texture as one moves away from this point. Connections between Wasserstein distortion and models of the human visual system are noted.

## 1 Introduction

Classical image compression algorithms are optimized to achieve high pixel-level fidelity between the source and the reconstruction. That is, one views images as vectors in Euclidean space and seeks to minimize the distance between the original and reproduction using metrics such as PSNR, SSIM (Wang et al., 2004), etc. (Avcıbaş et al., 2002; Dosselmann & Yang, 2005; Hore & Ziou, 2010). While effective to a large extent (Berger, 1971; Pearlman & Said, 2011; Sayood, 2017), these objectives have long been known to introduce artifacts, such as blurriness, into the reconstructed image (Wang & Bovik, 2009). Similar artifacts arise in image denoising (Buades et al., 2005), deblurring (Nah et al., 2021), and super-resolution (Kwon et al., 2015).

Recently, it has been observed that such artifacts can be reduced if one simultaneously maximizes the *realism*[1] of the reconstructed images. Specifically, one seeks to minimize the distance between some distribution induced by the reconstructed images and the corresponding distribution for natural images (Blau & Michaeli (2018); see also Delp & Mitchell (1991); Li et al. (2011); Saldi et al. (2014)). A reconstruction algorithm that ensures that these distributions are close will naturally be free of obvious artifacts; the two distributions cannot be close if one is supported on the space of crisp images and the other is supported on the space of blurry images. Image reconstruction under realism constraints has been a subject of intensive research of late, both of an experimental (Rippel & Bourdev, 2017; Tschannen et al., 2018; Agustsson et al., 2019; Mentzer et al., 2020) and theoretical (Klejsa et al., 2013; Blau & Michaeli, 2018; 2019; Matsumoto, 2018; 2019; Theis & Wagner, 2021; Chen et al., 2021; 2022; Wagner, 2022; Hamdi & Gündüz, 2023) nature.

Up to now, the dual objectives of fidelity and realism have been treated as distinct and even in tension (Blau & Michaeli, 2018; Zhang et al., 2021; Chen et al., 2022; Niu et al., 2023; Salehkalaibar et al., 2023). Yet they represent two attempts to capture the same notion, namely the differences perceived by a human observer. It is natural then to seek a simultaneous generalization of the two. Such a generalization could be more aligned with human perception than either objective alone, or even a linear combination of the two. The main contribution of this paper is one such generalization, *Wasserstein distortion*, which is grounded in models of the Human Visual System (HVS).

Realism objectives take several forms depending on how one induces a probability distribution from images. First, one can consider the distribution induced by the ensemble of full resolution images (Theis & Wagner, 2021; Theis et al., 2022; Wagner, 2022; Chen et al., 2022; Hamdi & Gündüz, 2023). Second, one can form a distribution over patches by selecting a patch at random from within a randomly selected image (Agustsson et al., 2019). Finally, for a given image, one can consider

---

[1]Realism is also referred to as *perceptual quality* by some authors.

the distribution over patches induced by selecting a location at random and extracting the resulting patch (Wang et al., 2018; Gao et al., 2021). Theoretical studies have tended to focus on the first approach while experimental studies have focused more on patches. We shall focus on the third approach because it lends itself more naturally to unification with fidelity: both depend only on the image under examination without reference to other images in the ensemble. That said, the proposed Wasserstein distortion can be extended naturally to videos and other sequences of images and in this way it generalizes the other notions of realism. Under an ergodicity assumption, as occurs with textures, ensemble and per-image notions of realism coincide; see Corollary C.4 to follow and the discussion in Portilla & Simoncelli (2000, p. 51).

Our simultaneous generalization of fidelity and realism is based in theories of the HVS, as noted above; namely it resorts to computing *summary statistics* in parts of the visual field where capacity is limited (Balas et al., 2009; Rosenholtz, 2011; Rosenholtz et al., 2012). In particular, Freeman & Simoncelli (2011) propose a model of the HVS focusing on the first two areas of the ventral stream, V1 and V2. The V1 responses are modeled as the outputs of oriented filters spanning the visual field with different orientations and spatial frequencies. The second area computes higher order statistics from the V1 outputs over various receptive fields. The receptive fields grow with eccentricity, as depicted in Fig. 1. In the visual periphery, the receptive fields are large and only the response statistics pooled over a large area are acquired. In the fovea, i.e., the center of gaze, the receptive field is assumed small enough that the statistics uniquely determine the image itself. See Freeman & Simoncelli (2011) for a complete description of the model. One virtue of this model is that it does not require separate theories of foveal and peripheral vision: the distinction between the two is simply the result of different receptive field sizes.

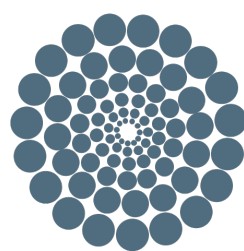

Figure 1: Receptive fields in the ventral stream grow with eccentricity.

This unification of foveal and peripheral vision likewise suggests a way of unifying fidelity and realism objectives. For each location in an image, we compute the distribution of features locally around that point using a weight function that decreases with increasing distance. The Wasserstein distance between the distributions computed for a particular location in two images measures the discrepancy between the images at that point. The overall distortion between the two images is then the sum of these Wasserstein distances across all locations. We call this *Wasserstein distortion*. If when constructing the distribution of features around a point, we use a strict notion of locality, i.e., a weight function that falls off quickly with increasing distance, then this reduces to a fidelity measure, akin to small receptive fields in Freeman and Simoncelli's model. If we use a loose notion of locality, i.e., a weight function that falls of slowly with distance, then this reduces a realism measure, akin to large receptive fields. Between the two is an intermediate regime with elements of both.

We propose the use of a one-parameter family of weight functions, where the parameter ($\sigma$) governs how strictly locality is defined. We find that to obtain good results requires careful selection of the family, especially its spectral properties. We prove that under a properly chosen weight function, Wasserstein distortion is a proper metric. In contrast, for the weighting function that is uniform over a neighborhood of variable size, which is popular in the texture generation literature, we exhibit adversarial examples of distinct pairs of images for which the distortion is zero.

The balance of the paper is organized as follows. Section 2 consists of a mathematical description of Wasserstein distortion. Section 3 discusses properties of the distortion measure, focusing in particular on the role of spectral properties of the weighting function. Section 4 contains our experimental results, specifically randomly generated images that are close to references under our distortion measure. All appendices represent supplementary materials.

## 2 DEFINITION OF WASSERSTEIN DISTORTION

We turn to defining Wasserstein distortion between a reference image, represented by a sequence $\mathbf{x} = \{x_n\}_{n=-\infty}^{\infty}$, and a reconstructed image, denoted by $\hat{\mathbf{x}} = \{\hat{x}_n\}_{n=-\infty}^{\infty}$. For notational simplicity, we shall consider 1-D sequences of infinite length, the 2-D case being a straightforward extension.

Let $T$ denote the unit advance operation, i.e., if $\mathbf{x}' = T\mathbf{x}$ then

$$x'_n = x_{n+1}. \tag{1}$$

We denote the $k$-fold composition $T \circ T \circ \cdots \circ T$ by $T^k$. Let $\phi(\mathbf{x}) : \mathbb{R}^{\mathbb{Z}} \mapsto \mathbb{R}^d$ denote a vector of local features of $\{x_n\}_{n=-\infty}^{\infty}$ about $n = 0$. The simplest example is the coordinate map, $\phi(\mathbf{x}) = x_0$. More generally, $\phi(\cdot)$ can take the form of a convolution with a kernel $\alpha(\cdot)$

$$\phi(\mathbf{x}) = \sum_{k=-m}^{m} \alpha(k) \cdot x_k, \tag{2}$$

or, since $\phi$ may be vector-valued, it can take the form of a convolution with several kernels of the form in (2). Following Portilla & Simoncelli (2000) and Freeman & Simoncelli (2011), one could choose $\phi(\cdot)$ to be a steerable pyramid (Simoncelli & Freeman, 1995; see also Balas et al., 2009; Rosenholtz, 2011; Rosenholtz et al., 2012). Following Ustyuzhaninov et al. (2017), the components of $\phi$ could take the form of convolution with a kernel as in (2), with random weights, followed by a nonlinear activation function. More generally, $\phi(\cdot)$ can take the form of a trained multi-layer convolutional neural network, as in Gatys et al. (2015).

Define the sequence $\mathbf{z}$ by

$$z_n = \phi(T^n \mathbf{x}) \tag{3}$$

and note that $z_n \in \mathbb{R}^d$ for each $n$. We view $\mathbf{z}$ as a representation of the image $\mathbf{x}$ in feature space.

Let $q_\sigma(k)$, $k \in \mathbb{Z}$, denote a family of probability mass functions (PMFs) over the integers, parameterized by $0 \leq \sigma < \infty$, satisfying:

**P.1** For any $\sigma$ and $k$, $q_\sigma(k) = q_\sigma(-k)$;

**P.2** For any $\sigma$ and $k, k' \in \mathbb{Z}$ such that $|k| \leq |k'|$, $q_\sigma(k) \geq q_\sigma(k')$;

**P.3** If $\sigma = 0$, $q_\sigma$ is the Kronecker delta function, i.e., $q_0(k) = \begin{cases} 1 & k = 0 \\ 0 & k \neq 0 \end{cases}$;

**P.4** For all $k$, $q_\sigma(k)$ is continuous in $\sigma$ at $\sigma = 0$;

**P.5** There exists $\epsilon > 0$ and $K$ so that for all $k$ such that $|k| \geq K$, $q_\sigma(k)$ is nondecreasing in $\sigma$ over the range $[0, \epsilon]$; and

**P.6** For any $k$, $\lim_{\sigma \to \infty} q_\sigma(k) = 0$.

We call $q_\sigma(\cdot)$ the *pooling PMF* and $\sigma$ the *pooling width* or *pooling parameter*. One PMF satisfying **P.1**-**P.6** is the *two-sided geometric distribution*,

$$q_\sigma(k) = \begin{cases} \frac{e^{1/\sigma}-1}{e^{1/\sigma}+1} \cdot e^{-|k|/\sigma} & \text{if } \sigma > 0 \\ 1 & \text{if } \sigma = 0 \text{ and } k = 0 \\ 0 & \text{otherwise.} \end{cases} \tag{4}$$

From the sequence $\mathbf{x}$, we define a sequence of probability measures $\mathbf{y}_\sigma = \{y_{n,\sigma}\}_{n=-\infty}^{\infty}$ via

$$y_{n,\sigma} = \sum_{k=-\infty}^{\infty} q_\sigma(k) \delta_{z_{n+k}}, \tag{5}$$

where $\mathbf{z}$ is related to $\mathbf{x}$ through (3) and $\delta_\cdot$ denotes the Dirac delta measure. Each measure $y_{n,\sigma}$ in the sequence represents the statistics of the features pooled across a region centered at $n$ with effective width $\sigma$. Note that all measures in $\mathbf{y}$ share the same countable support set in $\mathbb{R}^d$; they differ only in the probability that they assign to the points in this set. See Fig. 2. Similarly, we define $\hat{\mathbf{x}} = \{\hat{x}_n\}_{n=-\infty}^{\infty}$, $\hat{\mathbf{z}} = \{\hat{z}_n\}_{n=-\infty}^{\infty}$, and $\hat{\mathbf{y}}_\sigma = \{\hat{y}_{n,\sigma}\}_{n=-\infty}^{\infty}$ for the reconstructed image.

Let $d : \mathbb{R}^d \times \mathbb{R}^d \mapsto [0, \infty)$ denote an arbitrary distortion measure over the feature space. One natural choice is Euclidean distance

$$d(z, \hat{z}) = ||z - \hat{z}||_2, \tag{6}$$

although in general we do not even assume that $d$ is a metric. We define the distortion between the reference and reconstructed images at location $n$ to be

$$D_{n,\sigma} = W_p^p(y_{n,\sigma}, \hat{y}_{n,\sigma}), \tag{7}$$

where $W_p$ denotes the Wasserstein distance of order $p$ (Villani, 2009, Def. 6.1)[2]:

$$W_p(\rho, \hat{\rho}) = \inf_{Z \sim \rho, \hat{Z} \sim \hat{\rho}} \mathbb{E}\left[d^p(Z, \hat{Z})\right]^{1/p}, \tag{8}$$

where $\rho$ and $\hat{\rho}$ are probability measures on $\mathbb{R}^d$. The distortion over a block $\{-N, \ldots, N\}$ (such as a full image) is defined as the spatial average

$$D = D(\mathbf{x}, \mathbf{x}') = \frac{1}{2N+1} \sum_{n=-N}^{N} D_{n,\sigma}. \tag{9}$$

This assumes that the pooling parameter, $\sigma$, is the same for all $n$. In practice, it is desirable to vary the size of the pooling regions across the image. One can easily extend the above definition to allow $\sigma$ to depend on $n$:

$$D = D(\mathbf{x}, \mathbf{x}') = \frac{1}{2N+1} \sum_{n=-N}^{N} D_{n,\sigma(n)} = \frac{1}{2N+1} \sum_{n=-N}^{N} W_p^p\left(y_{n,\sigma(n)}, \hat{y}_{n,\sigma(n)}\right). \tag{10}$$

We call the function $\sigma(\cdot)$ the $\sigma$-*map*.

Wasserstein distance is widely employed due to its favorable theoretical properties, and indeed our theoretical results use the Wasserstein distance in (8) for some $p$ and $d$. In practice one might adopt a proxy for (8) that is easier to compute. Following the approach used with Fréchet Inception Distance (FID) (Heusel et al., 2017; Lucic et al., 2018; Liu et al., 2020; Fan et al., 2022), one could replace (8) with

$$||\mu - \hat{\mu}||_2^2 + \text{Tr}(C + \hat{C} - 2(\hat{C}^{1/2}C\hat{C}^{1/2})^{1/2}). \tag{11}$$

This is equivalent to $W_p^p$ if we take $p = 2$, $d$ to be Euclidean distance, and assume that $\rho$ (resp. $\hat{\rho}$) is Gaussian with mean $\mu$ (resp. $\hat{\mu}$) and covariance matrix $C$ (resp. $\hat{C}$) (Olkin & Pukelsheim, 1982). In our experiments, we simplify this even further by assuming that the features are uncorrelated,

$$\sum_{i=1}^{d} (\mu_i - \hat{\mu}_i)^2 + \left(\sqrt{V_i} - \sqrt{\hat{V}_i}\right)^2, \tag{12}$$

where $\mu_i$ and $V_i$ are the mean and variance of the $i$th component under $\rho$ and similarly for $\hat{\rho}$. This is justified when the feature set is overcomplete because the correlation between two features is likely to be captured by some third feature, as noted previously by Vacher et al. (2020). Other possible proxies include sliced Wasserstein distance (Pitié et al., 2005; Bonneel et al., 2015; Tartavel et al., 2016; Heitz et al., 2021), Sinkhorn distance (Cuturi, 2013), Maximum Mean Discrepancy (MMD) (Smola et al., 2006; Li et al., 2017; 2019), or the distance between Gram matrices (Gatys et al., 2015; Ustyuzhaninov et al., 2017).

The idea of measuring the discrepancy between images via the Wasserstein distance, or some proxy thereof, between distributions in feature space is not new (Rubner et al., 2000; Pitié et al., 2005; Tartavel et al., 2016; Vacher et al., 2020; Heitz et al., 2021; Elnekave & Weiss, 2022; Houdard et al., 2023). As they are concerned with ergodic textures or image stylization, these applications effectively assume a form of spatial homogeneity, which corresponds to the regime of large pooling regions ($\sigma \to \infty$) in our formulation, and empirical distributions with equal weights over the pixels. That is, the pooling PMF in (5) is taken to be uniform over a large interval centered at zero (e.g., Eq. (1) of Heitz et al. (2021)). Our goal here is to lift fidelity and realism into a common framework by considering the full range of $\sigma$ values, and we shall see next that for small or moderate values of $\sigma$, the uniform PMF is problematic.

## 3    PROPERTIES OF WASSERSTEIN DISTORTION

One can verify that as long as the pooling PMF satisfies **P.1**-**P.6**, as $\sigma \to 0$, Wasserstein distortion reduces to the $d$-distortion between $\mathbf{x}$ and $\hat{\mathbf{x}}$, raised to the power $p$ (Theorem C.1 in Appendix C).

---

[2]We refer to $W_p$ as the Wasserstein *distance* even though it is not necessarily a metric if $d$ is not a metric.

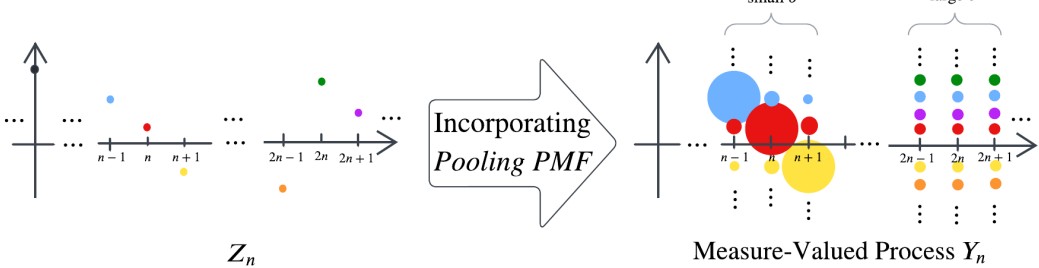

Figure 2: A pictorial illustration of (5). In the right plot, the size of the disk indicates the probability mass and the vertical coordinate of the center of the disk indicates the value.

Likewise, for ergodic processes $\mathbf{X}$ and $\hat{\mathbf{X}}$, as $\sigma \to \infty$, Wasserstein distortion reduces to the Wasserstein distance between the marginal distributions of $\mathbf{X}$ and $\hat{\mathbf{X}}$ (Theorem C.2 in Appendix C), again raised to the power $p$. Thus Wasserstein distortion subsumes fidelity and realism constraints and interpolates between them as desired.

In the $\sigma \to \infty$ regime, Wasserstein distortion will not be a true metric in that certain pairs of distinct $\mathbf{x}$ and $\mathbf{x}'$ will have zero distortion. Indeed, by Theorem C.2, this is true a.s. for a pair of realizations drawn independently from the same ergodic process. Practically speaking, when $\sigma$ is large, the Wasserstein distortion between two independent realizations of the same texture will be essentially zero (cf. Fig. 8 in Appendix B). When $\sigma$ is small, however, we want Wasserstein distortion to behave as a conventional distortion measure and as such it is desirable that it be a metric or a power thereof. In particular, we desire that it satisfy *positivity*, i.e., that $D(\mathbf{x}, \mathbf{x}') \geq 0$ with equality if and only if $\mathbf{x} = \mathbf{x}'$.

Whether Wasserstein distortion satisfies positivity at finite $\sigma$ depends crucially on the choice of the pooling PMF. Consider, for example, the popular uniform PMF:

$$q_m(k) = \begin{cases} \frac{1}{2m+1} & \text{if } |k| \leq m \\ 0 & \text{otherwise.} \end{cases} \tag{13}$$

In this case Wasserstein distortion does not satisfy positivity, even over the feature space, for any $m$. Let $D(\mathbf{z}, \mathbf{z}')$ denote Wasserstein distortion defined over the feature space, that is, without the composition with $\phi(\cdot)$. Observe that $D(\mathbf{z}, \mathbf{z}') = 0$ if $\mathbf{z}$ and $\mathbf{z}'$ are shifted versions of a sequence that is periodic with period $2m + 1$. If $m = 1$, for example, then the sequences

$$\mathbf{z} = \dots, a, b, c, a, b, c, a, b, c, \dots \tag{14}$$

$$\mathbf{z}' = \dots, b, c, a, b, c, a, b, c, a, \dots \tag{15}$$

satisfy $D(\mathbf{z}, \mathbf{z}') = 0$ because both $y_{n,\sigma}$ and $y'_{n,\sigma}$ are uniform distributions over $\{a, b, c\}$ for all $n$. See Fig. 3A for an example of distinct images for which the distortion is exactly zero assuming a uniform PMF and the coordinate feature map. In this case $D(\mathbf{x}, \mathbf{x}') = 0$ even if one uses the full Wasserstein distance in (8). If one uses a proxy, the situation is more severe. For MMD, for instance, the images in Fig. 3B have zero distortion at any $0 \leq \sigma < \infty$.

The problem lies with the spectrum of the pooling PMF. This is easiest to see in the case of MMD, for which the Wasserstein distortion reduces to the squared Euclidean distance between the convolution of the feature vectors with the pooling PMF. Thus if the pooling PMF has a spectral null, feature vectors that have all of their energy located at the null are indistinguishable from zero, which is how the adversarial examples in Fig. 3B were constructed. Conversely, if the pooling PMF has no spectral nulls, then Wasserstein distortion is the $1/p$-th power of a metric, as we show next. For this result, we assume that $\mathbf{x}$ and $\mathbf{x}'$ (resp. $\mathbf{z}$ and $\mathbf{z}'$) are finite-length sequences, and the indexing in (5) is wraparound.

**Theorem 3.1.** *For any $0 \leq \sigma < \infty$, if $d$ is a metric and $q_\sigma(\cdot)$ has no spectral nulls, then $D(\mathbf{z}, \mathbf{z}')^{1/p}$ is a metric. If, in addition, $\phi(\cdot)$ is invertible then $D(\mathbf{x}, \mathbf{x}')^{1/p}$ is also a metric.*

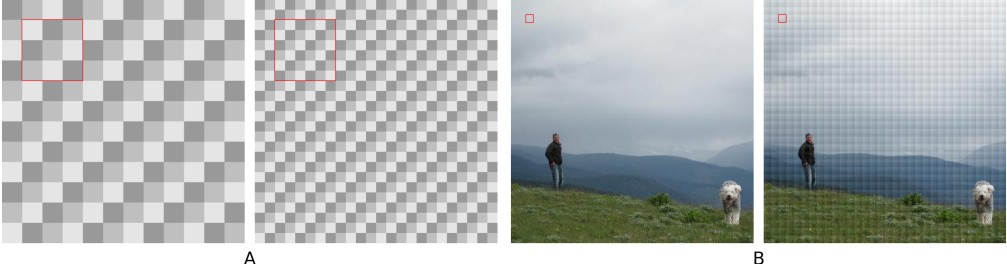

Figure 3: Examples showing that Wasserstein distortion does not satisfy positivity under a uniform PMF, where the red square in each image indicates the size of the pooling regions. The distortion between the two images on the left (A) is zero even if one uses the full Wasserstein distance in (5). If one uses MMD (Smola et al., 2006) as a proxy, then Wasserstein distortion with a uniform PMF is blind to certain blocking artifacts in that the two images on the right (B) have distortion zero. Compare Theorem 3.1. In both examples, $\phi(\cdot)$ is taken to be the coordinate map.

*Proof.* Since $d$ is a metric, we immediately have that $D(\cdot, \cdot)$ is symmetric, $D(\mathbf{z}, \mathbf{z}') \geq 0$, $D(\mathbf{z}, \mathbf{z}) = 0$ and similarly for $D(\mathbf{x}, \mathbf{x}')$. Suppose $\mathbf{z} \neq \mathbf{z}'$. Then since $q_\sigma(\cdot)$ has no spectral nulls, $q_\sigma * \mathbf{z} \neq q_\sigma * \mathbf{z}'$ (Oppenheim & Schafer, 1989, Eq. (8.120)), where $*$ denotes circular convolution. But if $u_n$ (resp. $u_n'$) denotes the mean of the measure $y_n$ (resp. $y_n'$), then $\mathbf{u} = q_\sigma * \mathbf{z}$ (resp. $\mathbf{u}' = q_\sigma * \mathbf{z}'$). Thus $\mathbf{y} \neq \mathbf{y}'$ since the sequence of means differ. It follows that $W_p^p(y_\ell, y_\ell') \neq 0$ for some $\ell$ and hence $D(\mathbf{z}, \mathbf{z}') > 0$ since $W_p$ is a metric (Villani, 2009, p. 94). If $\phi(\cdot)$ is invertible, then $\mathbf{x} \neq \mathbf{x}'$ implies $\mathbf{z} \neq \mathbf{z}'$, which implies $D(\mathbf{x}, \mathbf{x}') > 0$. That $D(\mathbf{x}, \mathbf{x}')$ and $D(\mathbf{z}, \mathbf{z}')$ satisfy the triangle inequality follows from the fact that $W_p$ is a metric and Minkowski's inequality. $\square$

When $\sigma$ is large, the PMF will be nearly flat over a wide range, so its spectrum will necessarily decay quickly. For small $\sigma$, the PMF is concentrated in time, so the spectrum can be made nearly flat in frequency if one chooses. Theoretically speaking, we need only to avoid PMFs with spectral nulls, such as the uniform distribution, to ensure positivity. Practically speaking, we desire pooling PMFs with a good *condition number*, meaning that the ratio of the maximum of the power spectrum to its minimum is small. In this vein, we note that the two-sided geometric PMF in (4) is well-conditioned, whereas the raised-cosine-type PMF used in (Freeman & Simoncelli, 2011, Eq. (9) with $t = 1/2$) has a condition number that is larger by almost four orders of magnitude for pooling regions around size 20. Note that papers in the literature that rely on uniform PMFs are focused on realism, i.e., the large $\sigma$ regime, for which the presence of spectral nulls is less of a concern.

## 4  Experiments

We validate Wasserstein distortion using the method espoused by Ding et al. (2021), namely by taking an image of random pixels and iteratively modifying it to reduce its Wasserstein distortion to given a reference image. Following Gatys et al. (2015), we use as our feature map selected activations within the VGG-19 network, with some modifications. See Appendix A for a description of VGG-19 and our modifications; here we note only that we augment the VGG-19 features to include the raw pixel values. We found that including this "0th" layer of the network provides for an improved reproduction of the DC level of the image. We use the scalar Gaussianized Wasserstein distance in (12) as a computational proxy for (8). For the pooling PMF, we take the horizontal and vertical offsets to be i.i.d. according to the two-sided geometric distribution in (4), conditioned on landing within the boundaries of the image. We minimize the Wasserstein distortion between the reference and reconstructed images using the L-BFGS algorithm (Zhu et al., 1997) with $4,000$ iterations and an early stopping criterion.

If the reference image is a texture and we take $\sigma$ to be large across the entire image, then we recover the standard texture generation setup (Gatys et al., 2015; Ustyuzhaninov et al., 2017; Heitz et al., 2021) The results are commensurate with dedicated texture synthesis schemes, which is unsurprising since with this $\sigma$-map, our setup is close to that of Heitz et al. (2021). See Appendix B for

sample images and further discussion. We focus here on experiments that demonstrate the utility of Wasserstein distortion's ability to interpolate between fidelity and realism.

## 4.1 EXPERIMENT 1: TRANSITING FROM FIDELITY TO REALISM

We consider generating a random image that is close to a challenging texture under Wasserstein distortion, sweeping $\sigma$ from zero to infinity. Specifically, $\sigma$ is constant across the image, but varies from run to run (Fig. 4). When $\sigma$ is close to zero, we recover the original image as expected. When $\sigma$ is large, we obtain an independent realization of the texture, again as expected. In between, we obtain images that balance both objectives. In particular, around $\sigma = 40$, individual pebbles can be associated between the original and the reconstruction, although they differ in their size, shape, orientation and markings.

The uniform PMF is often used in the literature, as noted earlier. One can perform the same experiment but using a uniform PMF over intervals of various widths. We find that the resulting progression from pure fidelity to pure realism is more abrupt, with a few images exhibiting intermediate behavior. This is especially true at lower resolution, for which the intermediate regime is quite narrow (Fig. 10 and Fig. 11 in Appendix B.2).

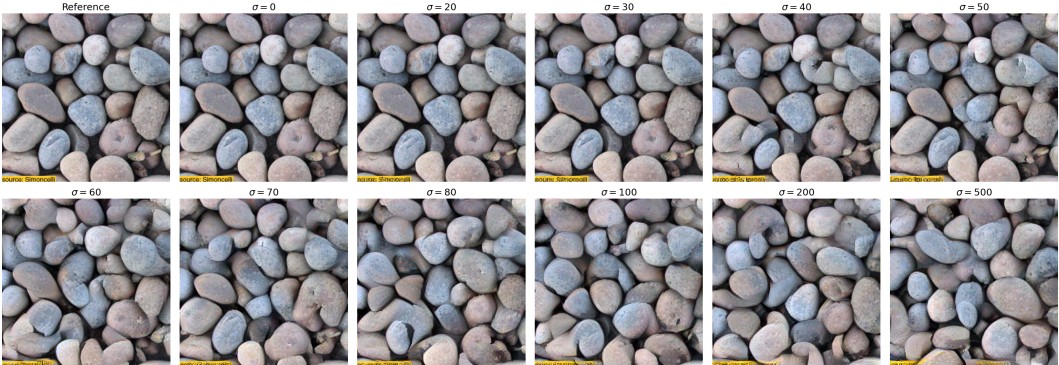

Figure 4: The first image is the reference; all others are reproductions under different $\sigma$'s. We see as $\sigma$ increases, the generated image transits from a pixel-accurate reproduction to an independent realization of the same texture.

## 4.2 EXPERIMENT 2: PINNED TEXTURE SYNTHESIS

We turn to an experiment in which $\sigma$ varies spatially over the image. Specifically, we consider a variation of the standard texture synthesis setup in which we set $\sigma = 0$ for pixels near the center; other pixels are assigned a $\sigma$ proportional to their distance to the nearest pixel with $\sigma = 0$, with the proportionality constant chosen so that the outermost pixels have a $\sigma$ that is comparable to the width of the image. The choice of having $\sigma$ grow linearly with distance to region of interest is supported by studies of the HVS, as described more fully in the next section. Under this $\sigma$-map, Wasserstein distortion behaves like a fidelity measure in the center of the image and a realism measure along the edges, with an interpolation of the two in between. The results are shown in Fig. 5 (see Fig. 8 in the supplementary material for additional examples). The $\sigma = 0$ points have the effect of pinning the reconstruction to the original in the center, with a gradual transition to an independent realization at the edge.

## 4.3 EXPERIMENT 3: REPRODUCTION OF NATURAL IMAGES WITH SALIENCY MAPS

Lastly, we consider natural images. We use the `SALICON` dataset (Jiang et al., 2015) which provides a saliency map for each image that we use to produce a $\sigma$-map. Specifically, we set a saliency threshold above which points are declared to be high salience. For such points we set $\sigma = 0$. For all other points $\sigma$ is proportional to the distance to the nearest high-salience point, with the proportionality constant determined by the constraint that the farthest points should have a $\sigma$ value on par with the width of the image. The choice of having $\sigma$ grow linearly with distance from the

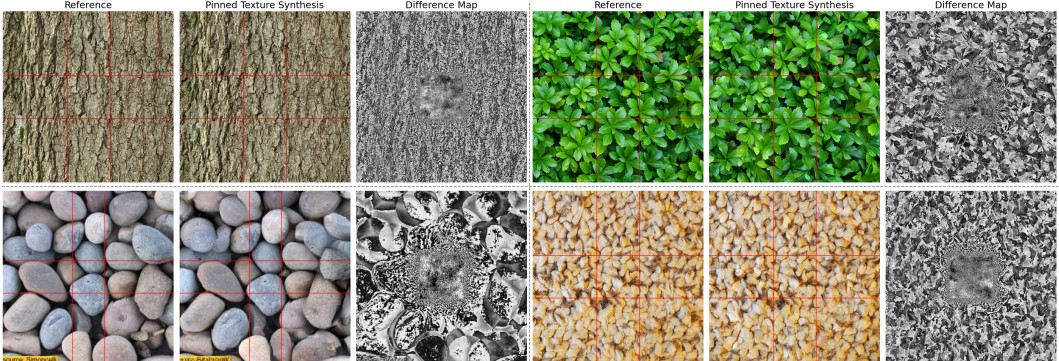

Figure 5: Examples from Experiment 2; auxiliary lines indicate the square of $\sigma = 0$ points at the center. The reconstructions smoothly transition from pixel-level fidelity at the center to realism at the edges.

high saliency region is supported by studies of the HVS. There is both physiological (Dumoulin & Wandell, 2008) and operational (Freeman & Simoncelli, 2011) evidence that the size of the receptive fields in the HVS grows linearly with eccentricity. If one seeks to produce images that are difficult for a human observer to easily distinguish, it is natural to match the pooling regions to the corresponding receptive fields when the gaze is focused on the high saliency region.

The results are shown in Fig. 6 (see also Fig. 9 in the supplementary materials). For images for which the non-salient regions are primarily textures, the reproductions are plausible replacements for the originals. In some other cases, the images appear to be plausible replacements if one focuses on high saliency regions, but not if one scrutinizes the entire image. This suggests that Wasserstein distortion can capture the discrepancy observed by a human viewer focused on high-saliency regions.

It should be emphasized that the process of producing the reconstructions in Figs. 6 and 9 requires no pre-processing or manual labeling. In particular, it is not necessary to segment the image. Given a binarized saliency map, the $\sigma$-map can be constructed automatically using the above procedure, at which point the Wasserstein distortion is well defined and training can begin.

## 5 DISCUSSION

Our present work lies on the intersection between models of the early human visual system, models of visual texture, and measures of both image realism and distortion.

We exploit a particularity of the HVS, which is its unique (among the various senses) ability to foveate, and hence extract information preferentially from spatial locations selected by gaze. In this regard, our work most directly leans on that of Freeman & Simoncelli (2011), but also has clear connections to Balas et al. (2009); Rosenholtz (2011); Rosenholtz et al. (2012), who consider a *summary statistics* model of the visual periphery. However, as these studies mainly aim to explain the HVS, their focus is not to provide a unified, optimizable metric in the mathematical sense, as provided in the present work. Wasserstein distortion can *quantify* how far an image is from a metamer, whereas Freeman & Simoncelli (2011) cannot.

Texture generation as an image processing tool is closely tied to the notion of spatial ergodicity, and our work finds itself in a long line of probabilistic models built on this assumption (e.g., Chellappa & Kashyap, 1985; Heeger & Bergen, 1995; Efros & Leung, 1999; Portilla & Simoncelli, 2000; Kwatra et al., 2005; Gatys et al., 2015). To our knowledge, the notion of capturing spatial correlations of pixels not directly, but by considering simple, mathematically tractable statistics in potentially complex feature spaces, traces back to Zhu et al. (1998). Like Freeman & Simoncelli (2011), our work combines this notion with the spatial adaptivity of the HVS, but is mathematically much more concise. Our use of a Wasserstein divergence in this particular context is predated by Vacher et al. (2020) and others, whose work is however limited to ergodic textures.

| Reference | Reproduction of Natural Images | Difference Map | Saliency Map |
|---|---|---|---|

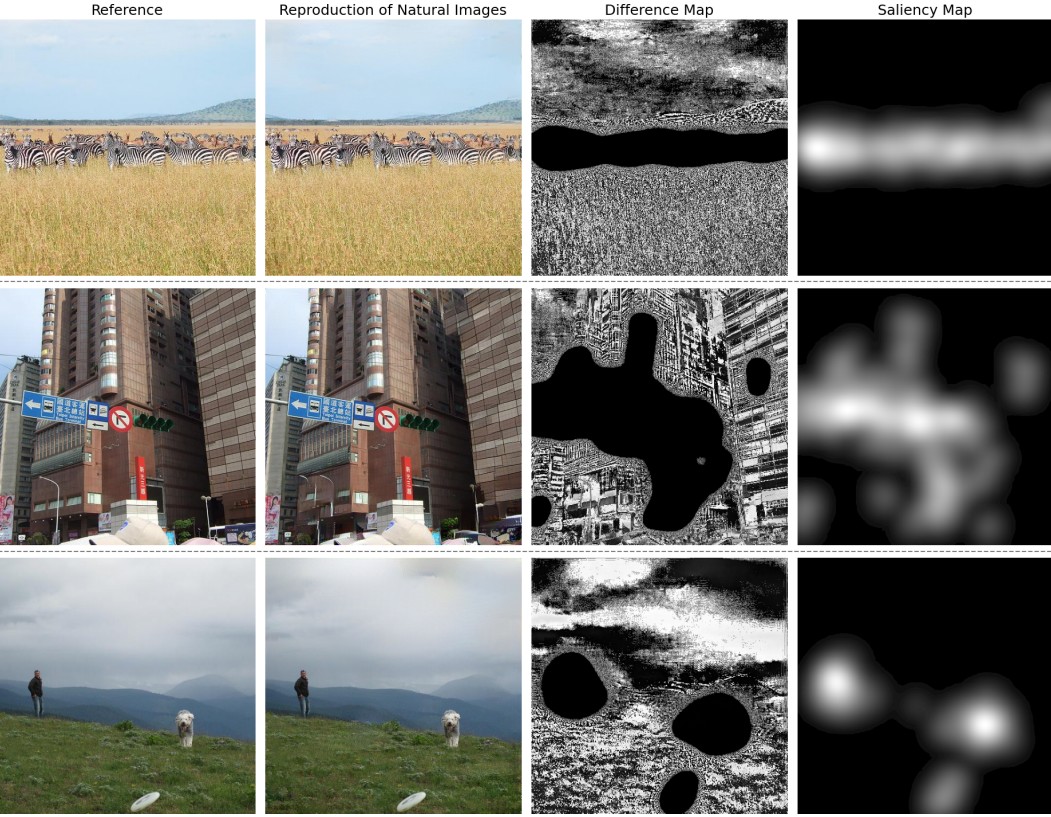

Figure 6: For each row, the first image is the reference image and the second is the reproduction; the third is the difference between the two; and the fourth is the saliency map from SALICON before binarization. In the high saliency regions, the reconstruction exhibits pixel-level fidelity. Elsewhere, it exhibits realism or an interpolation of the two. Note that the goal of this experiment is not to reproduce images that withstand visual scrutiny in all regions, but to demonstrate how Wasserstein distortion becomes increasingly permissive to error towards the visual periphery, and that the errors that are permitted can be quite difficult to spot when viewing the salient regions at an appropriate distance. Compare with Freeman & Simoncelli (2011, Fig. 2); additional examples in Fig. 9 (supplement).

As a measure of realism, Wasserstein distortion is related to the Fréchet inception distance (Heusel et al., 2017), which, as our experimental results do, uses a Gaussianized Wasserstein divergence in a feature space induced by pretrained neural networks. However, the FID is a measure of realism across the ensemble of images, rather than across space. In our view, the concept of realism as a divergence across ensembles of full-resolution images is at odds with the everyday observation that humans can distinguish realistic from unrealistic images by looking at a single example. Wasserstein distortion offers one possible explanation for *how* humans might make these one-shot judgements, namely by measuring realism across spatial regions. The HVS studies mentioned above support this notion. Spatial realism may play a crucial role in modeling human perception, in particular in the visual periphery; and hence, for all practical applications, in regions of low saliency.

In terms of future work, the application of Wasserstein distortion to compression is natural and as yet unexplored. Practical image compressors optimized for Wasserstein distortion could encode statistics over pooling regions that vary in size depending on the distance from the salient parts of the image. Note that this approach would be distinct from only encoding high-saliency regions and using a generative model optimized for ensemble realism to "fill in" the remainder. The latter approach would rely on knowledge of the conditional distribution given the encoding rather than the local image statistics. As such, it would be allowed to deviate more significantly from the source image, so long as low-saliency regions that it creates are contextually plausible.

REPRODUCIBILITY STATEMENT

For the theoretical results, all assumptions are stated in or immediately before the theorems, and self-contained proofs are provided in the same section as the theorems.

For the experiments, the source code for Experiment 2 in Section 4.2, along with some reference-reproduction sample pairs, are provided as supplementary material. The code is self-contained, and tested on Ubuntu 22.04 LTS with Python 3.8.10 and Python 3.10.12 with TensorFlow 2.11.0 and CUDA GPUs. To run the code, execute `wass_dist_texture_synthesis.py`.

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

## A    EXPERIMENTAL SETUP

Our work utilizes the VGG-19 network, although we emphasize that the framework is agnostic to the choice of features. Details of the VGG-19 network can be found in Simonyan & Zisserman (2015); we use the activation of selected layers as our features, with the following changes to the network structure:

1. All pooling layers in the original network in Simonyan & Zisserman (2015) use MaxPool; as suggested by Gatys et al. (2015), in our experiments we use AvePool;

2. There are 3 fully connected layers and a soft-max layer at the end in the original structure in Simonyan & Zisserman (2015), which we do not use;

3. We use the weights pre-trained on ImageNet (Deng et al., 2009), that are normalized such that over the validation set of ImageNet, the average activation of each layer is 1, as suggested in Gatys et al. (2015).

Fig. 7 provides a illustration.

The Wasserstein distortion is defined at every location in the image. This is entirely analogous to the way that the squared error between images is defined at every location. In our case, computational limitations prevent us from evaluating the distortion at every location in images of a reasonable size unless $\sigma$ is small. In parts of the image in which $\sigma$ is large, we found that satisfactory results could be obtained by evaluating the distortion at a subset of points, with the subset randomly selected between iterations. When $\sigma = 0$, Wasserstein distortion reduces to MSE, and in this case, we skip the computation of the variance in (12), since it must be zero. This affords some reduction in computation, which allows us to evaluate the distortion at a larger set of points. The locations at which distortion is computed are called *pixels of interest.*

For all our experiments, the Wasserstein distortion is calculated as follows: we pass the pre-processed source image $\mathbf{x}$ and reconstruction image $\hat{\mathbf{x}}$ through the VGG-19 network, and denote the response activation of each layer $\ell$ by $\mathbf{z}^\ell$ and $\hat{\mathbf{z}}^\ell$, respectively. We denote the source and reconstruction image themselves as the 0th layer. In pre-processing, we do not remove the DC component of the image, in contrast to the training process of VGG-19 network. For each experiment, we specify

1. a set of layers of interest;

2. for each spatial dimension, a method to compute the $\sigma$-map;

3. a method to determine the pixel of interest;

4. a multiplier $M_\ell$ and $M_\sigma$ for each layer $\ell$ and each $\sigma$, respectively.

The activation response of all layers of interest (with activation being the identity map for 0-th layer if it is one layer of interest) can be seen as the feature $\phi(\mathbf{x})$ in our construction. For each layer of interest $\ell$, we obtain the sequences of probability measures $\mathbf{y}^\ell$ and $\hat{\mathbf{y}}^\ell$ from $\mathbf{z}^\ell$ and $\hat{\mathbf{z}}^\ell$; for each pixel of interest $(i, j)^\ell$ in layer $\ell$, we calculate the Wasserstein distortion $D^\ell_{i,j,\sigma}(y^\ell_{i,j,\sigma}, \hat{y}^\ell_{i,j,\sigma}) \times M_\sigma \times M_\ell$ with (12), where the $\sigma$ is determined by the $\sigma$-map. The loss is

$$D = \sum_\ell \sum_{(i,j)^\ell} D^\ell_{i,j,\sigma}(y^\ell_{i,j,\sigma}, \hat{y}^\ell_{i,j,\sigma}) \times M_\sigma \times M_\ell. \tag{16}$$

For the first experiment, we use all layers up to (but excluding) the 4th pooling layer (`pool4` in Fig. 7), and the weight is the inverse of the normalization factor; effectively raw ImageNet weights are applied.

For the second experiment, we use all layers up to (but excluding) the 4th pooling layer (`pool4` in Fig. 7). For each layer, we evaluate the distortion at all high fidelity ($\sigma = 0$) pixels and 25 randomly chosen pixels that are not high fidelity pixels. We randomly choose 20 sets of 25 pixels, and randomly use one of the sets in each distortion calculation. For the 0th layer, $M_{\ell=0} = 100$; for the first $1/3$ layers, $M_\ell = 10$; for the middle $1/3$ layers, $M_\ell = 5$; and for the last $1/3$ layers, $M_\ell = 1$. $M_{\sigma=0} = 1$ and $M_{\sigma \neq 0} = 200$.

The setup for the third experiment differs from the second only in the choice of $\sigma$ map. From the given saliency map, we obtain a "saliency value" for each pixel, normalized to $[0, 1]$. All pixels with saliency value higher than the threshold of $0.1$ are declared high saliency. The $\sigma$ values for the remaining locations are determined using the procedure described in the main text.

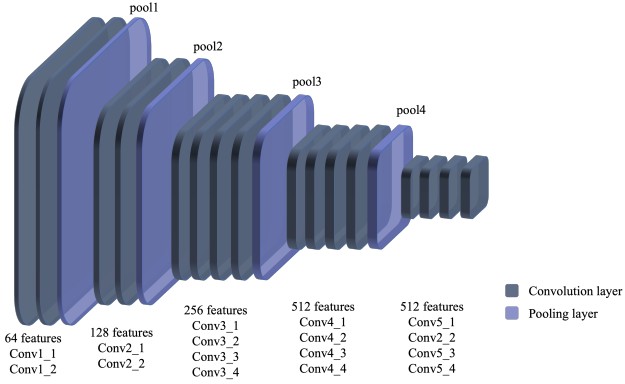

Figure 7: VGG-19 network structure.

## B FURTHER EXPERIMENTAL RESULTS

Fig. 8 exhibits more samples from Experiments 1 and 2, with $\sigma$ set to a very large value for Experiment 1. Fig. 9 exhibits more samples from Experiment 3. We note that runtime for Experiment 3 with a $480\times480$ reference image is 3 hours on average on an NVidia GTX4090 GPU.

### B.1 INDEPENDENT TEXTURE SYNTHESIS

We consider the canonical problem of generating an independent realization of a given texture (Portilla & Simoncelli, 2000; Gatys et al., 2015; Ustyuzhaninov et al., 2017; Heitz et al., 2021). We evaluate the Wasserstein distortion at a single point in the center of the image with $\sigma = 4,000$. Since the images are 256x256 or 512x512, the Wasserstein distortion effectively acts as a realism objective, per Theorem C.2.

The results are shown in the first two columns of Fig. 8. The results are commensurate with dedicated texture synthesis schemes (Gatys et al., 2015; Ustyuzhaninov et al., 2017; Heitz et al., 2021), which is unsurprising since with this $\sigma$-map, our setup is close to that of Heitz et al. (2021) as noted in the main text. The primary difference is that we use the 1-D Gaussianized Wasserstein distance in (12) in place of the sliced Wasserstein distance, which affords some computational savings. If there are $d$ features within a layer and $N$ pixels, the complexity of the scalar Gaussianized Wasserstein distance is $dN$ compared with $d^2 N + dN \log N$ for sliced Wasserstein distance (assuming $d$ random projections, as is done in Heitz et al. (2021)). In practice, we find that this translates to a speedup of about 2x, with comparable quality on the textures of interest. We conclude that, at least with VGG-19 and the textures considered here, it is unnecessary to compute the full 1-D Wasserstein distance along random directions; comparing the first two moments along the coordinate axes is sufficient. We note again, however, that the framework is agnostic to the choice of metric and sliced Wasserstein distance can be accommodated equally well.

### B.2 $\sigma$-PROGRESSION

We consider a variation of the experiment in Section 4.1: we downsample the pebble texture image, and generate random images under Wasserstein distortion with various $\sigma$'s. As a comparison, we repeat the experiment with the pooling PMF replaced by uniform weighting over patches of varying widths. The sequence of reproductions using a two-sided geometric PMF is in Fig. 10, and the sequence of reproductions under uniform weighting is in Fig. 11. We find that reproductions under the nonuniform PMF achieves a smooth transit from the reference to an independent realization, but

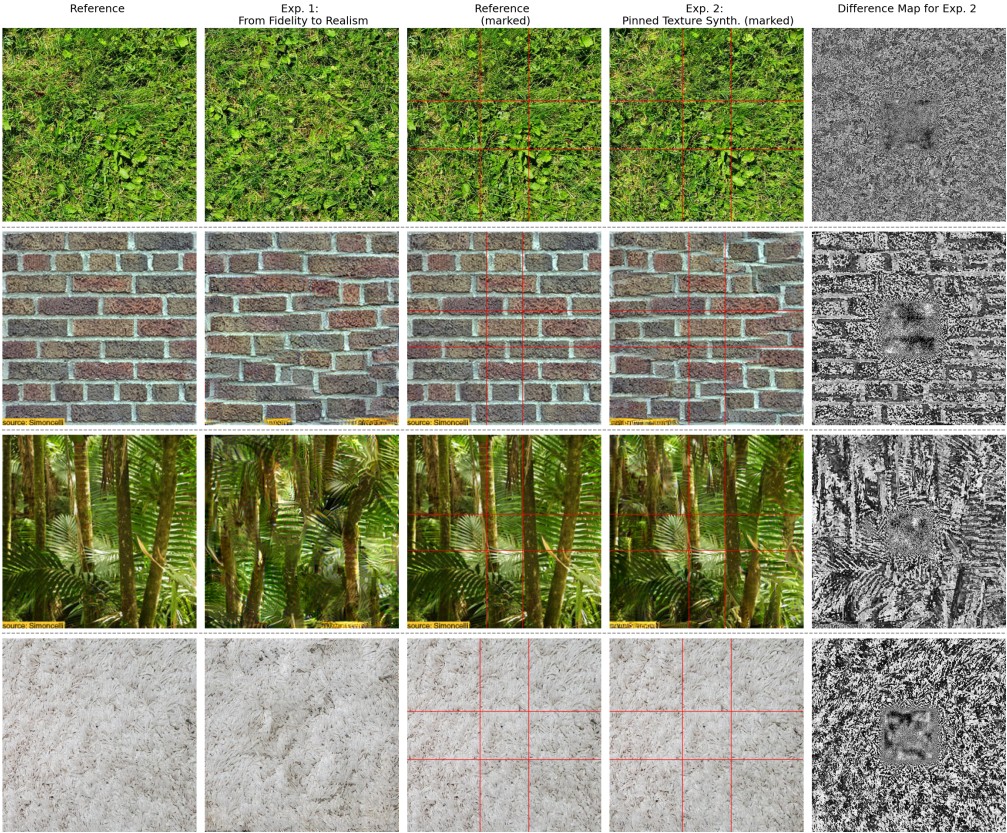

Figure 8: An extension of Fig. 5. For each row, the first two images are the reference-reproduction pair under Experiment 1; the results are commensurate with dedicated texture generators. The third and fourth images are the reference-reproduction pair under Experiment 2, with auxiliary lines drawn indicates the $\sigma = 0$ points; the fifth image is the difference of reference and reproduction textures under Experiment 2.

the reproductions under uniform weighting abruptly changes from the former to the latter when the width changes from 130 to 132 pixels.

## B.3  WASSERSTEIN DISTORTION VALUES FOR TEXTURES

Unless $\sigma$ is small, we do not expect the Wasserstein distortion between images to be low if and only if the images are identical. Rather, it should be small if and only if the perceptual differences between the two are minor. To validate this hypothesis, we calculate the Wasserstein distortion between a variety of textures. Results are shown in Fig. 12. All pixels are assigned $\sigma = 4,000$, with 9 pixels of interest that forms an even grid. Using (12) as the distortion measure, so long as the $\sigma$ maps and sets of pixels of interest are compatible, we can compute the Wasserstein distortion between two images even they have different resolutions. We see that the distortion is small for images of the same texture and large for images that represent different textures.

## B.4  SEPARATE FIDELITY AND REALISM CONSTRAINTS ARE INSUFFICIENT

We illustrate that a linear combination of MSE distortion and Gram matrix distortion as defined in Gatys et al. (2015); Ustyuzhaninov et al. (2017), cannot achieve the reconstruction obtained through Wasserstein distortion. We calculate the MSE distortion over the high saliency area defined in Section 4.3, the Gram matrix distortion over the whole image as defined in Gatys et al. (2015); Ustyuzhaninov et al. (2017), and linearly combine the two with a multiplier on the Gram matrix distortion. Fig. 13 exhibits the results.

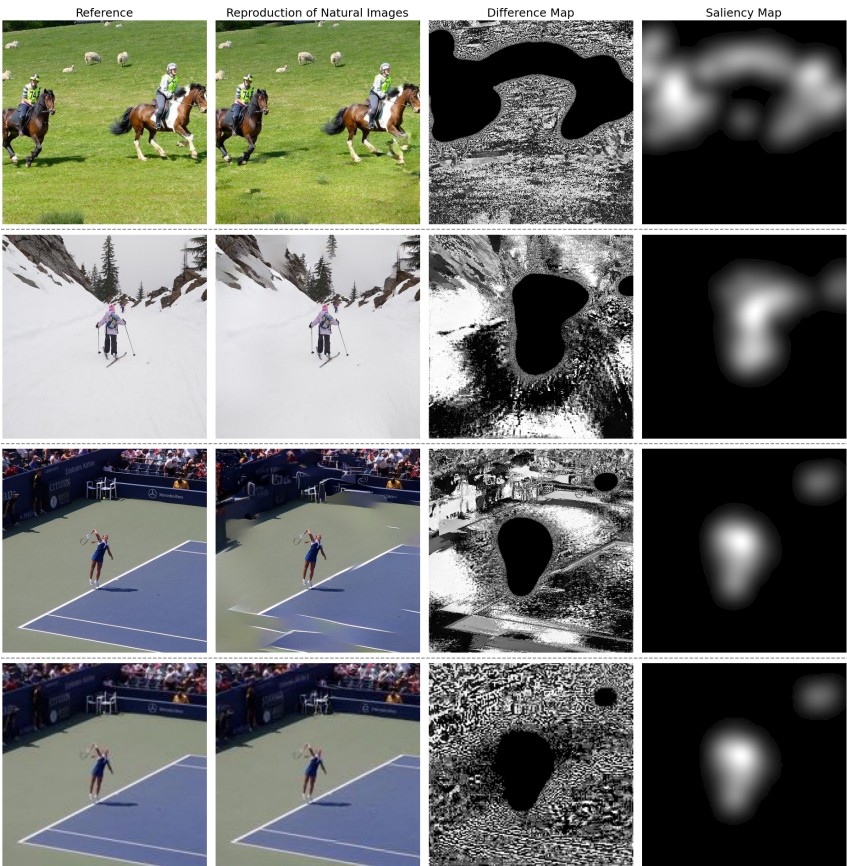

Figure 9: An extension of Fig. 6, consisting of more examples from Experiment 3. The misplaced foul lines in the third example are likely a manifestation of VGG-19's recognized difficulty with reproducing long linear features in textures (Liu et al., 2016; Snelgrove, 2017; Sendik & Cohen-Or, 2017; Zhou et al., 2018; Gonthier et al., 2022). This is evidenced through the fourth example, where the reference image has been downsampled so that VGG-19 better captures the long-range dependence.

## B.5 COMPARISON TO OTHER METRICS

We compare Wasserstein distortion to sliced Wasserstein loss proposed in Heitz et al. (2021). We applied both metrics to the independent texture synthesis task depicted in Appendix B.1. The results are shown in Fig. 14.

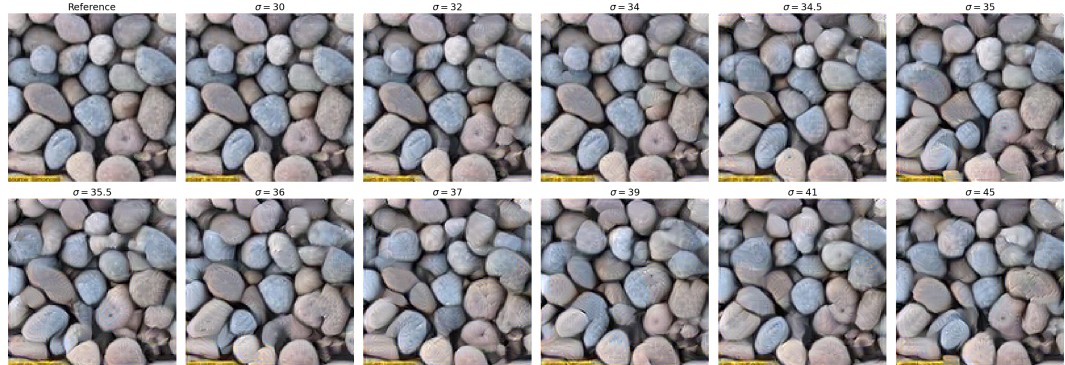

Figure 10: $\sigma$-progression for downsampled pebble image. The first image is the reference. The reproduction begins to shift away from the reference at $\sigma = 32$, and gradually converges to an independent realization an $\sigma$ grows.

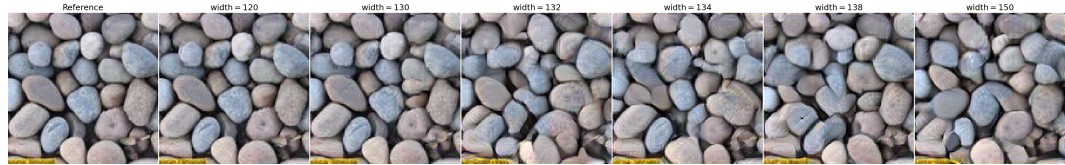

Figure 11: Similar to the previous figure but with a uniform pooling PMF with various widths. The reproduction moves from a high-fidelity recovery of the reference to an independent realization rather abruptly as the width of the pooling region moves from 130 to 132 pixels.

## C  FIDELITY AND REALISM AS EXTREME CASES

Let $\mathbf{x}$ and $\hat{\mathbf{x}}$ be two sequences and let $\mathbf{z}$ and $\hat{\mathbf{z}}$ denote the associated feature sequences, i.e., $z_n = \phi(T^n \mathbf{x})$ and $\hat{z}_n = \phi(T^n \hat{\mathbf{x}})$. If one is only concerned with fidelity to the original image, one might use an objective such as

$$\frac{1}{2N+1} \sum_{n=-N}^{N} d^p(z_n, \hat{z}_n), \tag{17}$$

perhaps with $\phi$ being the identity map; conventional mean squared error can be expressed in this way. This objective can be trivially recovered from Wasserstein distortion by taking $\sigma = 0$, invoking **P.3**, and applying the formula for the Wasserstein distance between point masses:

$$W_p(\delta_z, \delta_{\hat{z}}) = d(z, \hat{z}). \tag{18}$$

Given that we are interested in smoothly interpolating between fidelity and realism, we would like Wasserstein distortion to reduce to (17) in the limit as $\sigma \to 0$. We next identify conditions under which this continuity result holds. Note that this result does not require $d$ to be a metric.

**Theorem C.1.** *Suppose $q$ satisfies **P.3** – **P.5** and $\mathbf{z}$, $\hat{\mathbf{z}}$, and $q$ together satisfy*

$$\sum_{k=-\infty}^{\infty} q_\sigma(k) d^p(z_k, \hat{z}_k) < \infty \tag{19}$$

*for all $\sigma > 0$. Then we have*

$$\lim_{\sigma \to 0} D_{0,\sigma} = d^p(z_0, \hat{z}_0). \tag{20}$$

*Proof.* Fix $K$ and $\epsilon$ as in **P.5**. Consider the coupling between $y_{0,\sigma}$ and $\hat{y}_{0,\sigma}$ suggested by the ordering of the sequences:

$$D_{0,\sigma} = \inf_{Z \sim y_{n,\sigma}, \hat{Z} \sim \hat{y}_{n,\sigma}} E[d^p(Z, \hat{Z})] \leq \sum_{k=-\infty}^{\infty} q_\sigma(k) d^p(z_k, \hat{z}_k). \tag{21}$$

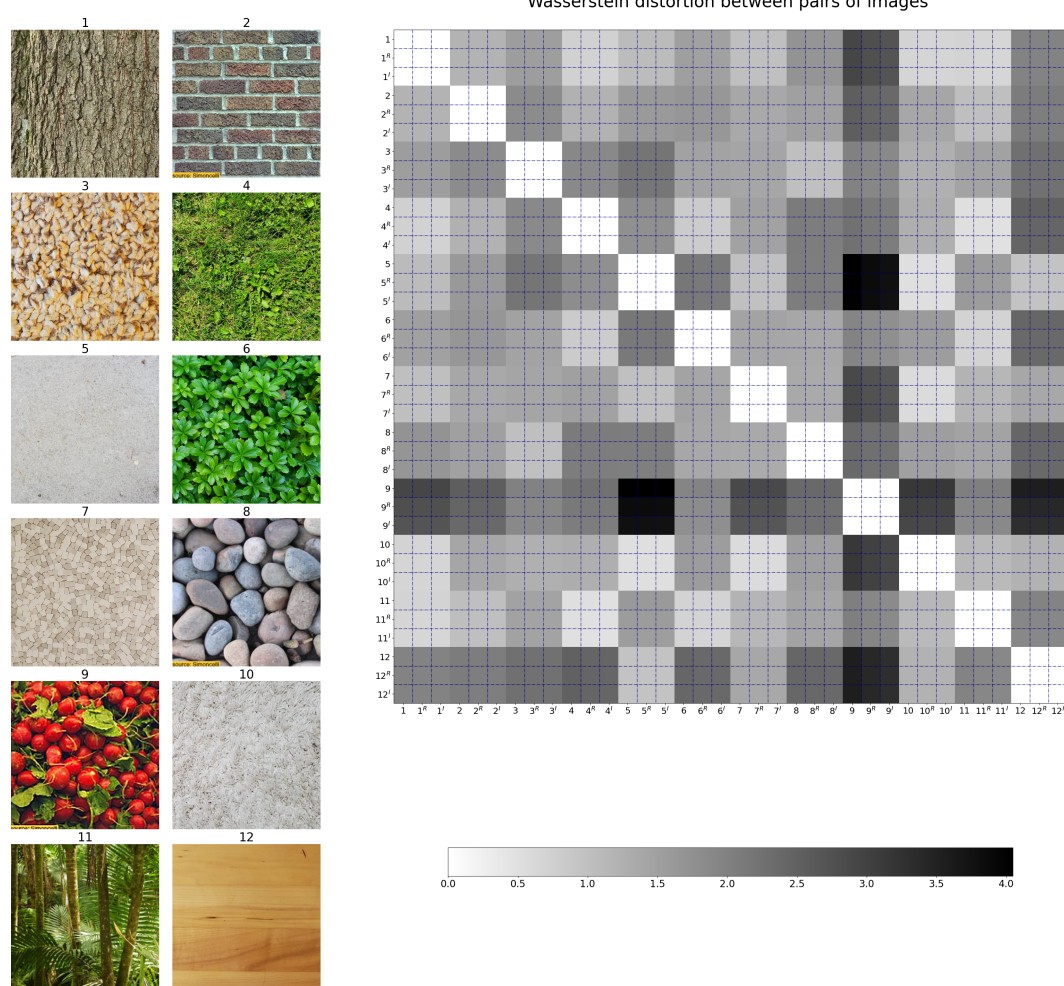

Figure 12: Wasserstein distortion between pairs of textures, normalized by the number of features and the number of pixels of interest. Each `number` refers to one reference texture; $\text{number}^R$ refers to the corresponding pinned reproduction texture (see Fig. 5 and Fig. 8), and $\text{number}^I$ refers to the corresponding independent reproduction texture (see Fig. 8). We see that the Wasserstein distortion between realizations of the same texture are small compared with the Wasserstein distortion between different textures.

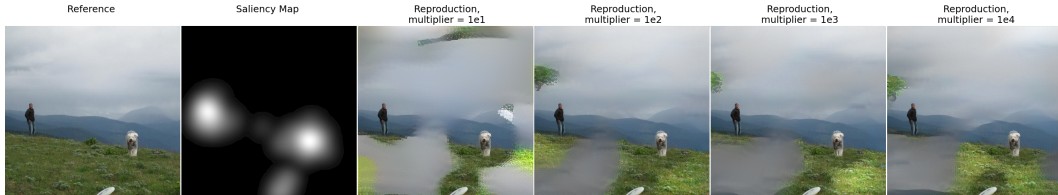

Figure 13: The first two images are the reference image and its saliency map; the rest are reproductions, with different multipliers applied to the Gram matrix distortion in the linear combination.

We have

$$\limsup_{\sigma \to 0} D_{0,\sigma} \leq \lim_{\sigma \to 0} q_\sigma(0) d^p(z_0, \hat{z}_0) + \lim_{\sigma \to 0} \sum_{k:0<|k|\leq K} q_\sigma(k) d^p(z_k, \hat{z}_k) \tag{22}$$

$$+ \lim_{\sigma \to 0} \sum_{k:|k|>K} q_\sigma(k) d^p(z_k, \hat{z}_k)$$

$$= d^p(z_0, \hat{z}_0), \tag{23}$$

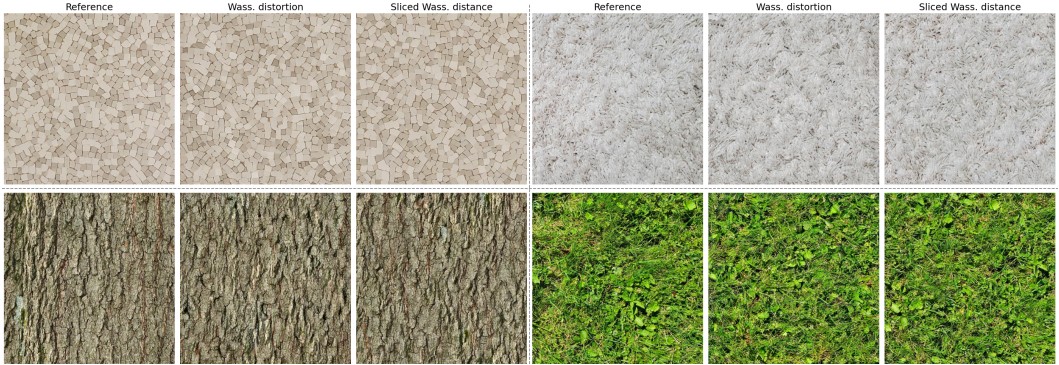

Figure 14: Texture synthesis task as described in Appendix B.1, under Wasserstein distortion and sliced Wasserstein distance. We are overall 2x faster (see discussion on computation complexity in Appendix B.1), with comparable reproduction images; also, using sliced Wasserstein distance would require initiating from the average color of the reference, while there is no restriction for Wasserstein distortion. We conclude that using Wasserstein distortion would be more efficient, with comparable synthesis quality and significantly reduced computation time.

where (23) follows from **P.3** and **P.4** (for the first two limits) and from **P.3**-**P.5** and dominated convergence (for the third limit). For the reverse direction, fix $\sigma > 0$ and let $q_\sigma(\cdot, \cdot)$ denote any PMF over $\mathbb{Z}^2$, both of whose marginals are $q_\sigma(\cdot)$. Then we have

$$\sum_{k_1=-\infty}^{\infty} \sum_{k_2=-\infty}^{\infty} q_\sigma(k_1, k_2) d^p(z_{k_1}, \hat{z}_{k_2}) \geq q_\sigma(0, 0) d^p(z_0, \hat{z}_0) \tag{24}$$

$$\geq (2 q_\sigma(0) - 1) d^p(z_0, \hat{z}_0), \tag{25}$$

from which the result follows by **P.3** and **P.4**. $\qquad\square$

Likewise, we show that Wasserstein distortion continuously reduces to pure realism in the large-$\sigma$ limit. We use $\overset{w}{\to}$ to denote weak convergence.

**Theorem C.2.** *Suppose $q$ satisfies P.1, P.2, and P.6 and $d$ is a metric. Let $F_N$ (resp. $\hat{F}_N$) denote the empirical CDF of $\{z_{-N}, \ldots, z_N\}$ (resp. $\{\hat{z}_{-N}, \ldots, \hat{z}_N\}$) and suppose we have*

$$F_N \overset{w}{\to} F \text{ and } \hat{F}_N \overset{w}{\to} \hat{F} \tag{26}$$

$$\int d^p(z, 0) dF_N \to \int d^p(z, 0) dF < \infty \text{ and } \int d^p(z, 0) d\hat{F}_N \to \int d^p(z, 0) d\hat{F} < \infty \tag{27}$$

*and, for all $\sigma$,*

$$\sum_{k=-\infty}^{\infty} q_\sigma(k) d^p(z_k, 0) < \infty \text{ and } \sum_{k=-\infty}^{\infty} q_\sigma(k) d^p(\hat{z}_k, 0) < \infty. \tag{28}$$

*Then we have*

$$\lim_{\sigma \to \infty} D_{0,\sigma} = W_p^p(F, \hat{F}). \tag{29}$$

To prove Theorem C.2, we need a lemma first.

**Lemma C.3** (Equivalence of Cesàro Sums). *Suppose $q$ satisfies P.1, P.2, and P.6. For any two-sided $\mathbb{R}$-valued sequence $\mathbf{a}$, if*

$$\lim_{m \to \infty} \frac{1}{2m+1} \sum_{k=-m}^{m} a_k = \alpha \in \mathbb{R}, \tag{30}$$

*and for all $\sigma > 0$,*

$$\sum_{k=-\infty}^{\infty} q_\sigma(k)|a_k| < \infty, \tag{31}$$

*then we likewise have*

$$\lim_{\sigma \to \infty} \sum_{k=-\infty}^{\infty} q_\sigma(k)a_k = \alpha. \tag{32}$$

*Proof.* We can write

$$\sum_{\ell=-\infty}^{\infty} q_\sigma(\ell)a_\ell = \lim_{k \to \infty} \sum_{\ell=-(k-1)}^{k-1} \left(q_\sigma(\ell) - q_\sigma(k)\right) a_\ell \tag{33}$$

$$= \lim_{k \to \infty} \sum_{\ell=-(k-1)}^{k-1} \sum_{m=|\ell|}^{k-1} \left(q_\sigma(m) - q_\sigma(m+1)\right) a_\ell \tag{34}$$

$$= \lim_{k \to \infty} \sum_{m=0}^{k-1} \sum_{\ell=-m}^{m} \left(q_\sigma(m) - q_\sigma(m+1)\right) a_\ell \tag{35}$$

$$= \sum_{m=0}^{\infty} \sum_{\ell=-m}^{m} \left(q_\sigma(m) - q_\sigma(m+1)\right) a_\ell, \tag{36}$$

where (33) holds by (31), (**P.1**), (**P.2**), and dominated convergence. For $m \geq 0$, define the sequences

$$b_m = \frac{1}{2m+1} \sum_{\ell=-m}^{m} a_\ell \tag{37}$$

and

$$r_\sigma(m) = \left(q_\sigma(m) - q_\sigma(m+1)\right)(2m+1). \tag{38}$$

By (**P.2**), $r_\sigma(m) \geq 0$. By (36),

$$\sum_{\ell=-\infty}^{\infty} q_\sigma(\ell)a_\ell = \sum_{m=0}^{\infty} r_\sigma(m)b_m. \tag{39}$$

Now the choice $a_\ell = 1$ satisfies (31) and in this case the previous equation reads $\sum_{m=0}^{\infty} r_\sigma(m) = 1$. Fix $\epsilon > 0$ and $M$ such that for all $m > M$, $|b_m - \alpha| < \epsilon$. We can write

$$\left| \sum_{\ell=-\infty}^{\infty} q_\sigma(\ell)a_\ell - \alpha \right| \leq \left| \sum_{m=0}^{M} r_\sigma(m)(b_m - \alpha) \right| + \left| \sum_{m=M+1}^{\infty} r_\sigma(m)(b_m - \alpha) \right| \tag{40}$$

$$\leq \left( \sum_{m=0}^{M} r_\sigma(m) \right) \left( \max_{m=0,1,\ldots,M} b_m + |\alpha| \right) + \epsilon. \tag{41}$$

Taking $\sigma \to \infty$ on both sides, the conclusion follows by **P.6**. $\qquad \square$

We now prove Theorem C.2.

*Proof of Theorem C.2.* With a slight abuse of notation we let $F_\sigma$ denote the CDF of the distribution

$$\sum_{k=-\infty}^{\infty} q_\sigma(k)\delta_{z_k} \tag{42}$$

and define $\hat{F}_\sigma$ analogously. Then $D_{0,\sigma} = W_p^p(F_\sigma, \hat{F}_\sigma)$. By the triangle inequality for Wasserstein distance (Villani, 2009, p. 94) (which requires $d$ to be a metric),

$$W_p(F_\sigma, \hat{F}_\sigma) \leq W_p(F_\sigma, F) + W_p(F, \hat{F}) + W_p(\hat{F}, \hat{F}_\sigma). \tag{43}$$

By Lemma C.3 and (26), $F_\sigma \xrightarrow{w} F$. By Lemma C.3, (27), and (28), we have

$$\lim_{\sigma \to \infty} \int d^p(z, 0) dF_\sigma(z) = \int d^p(z, 0) dF(z). \tag{44}$$

These two conditions imply that $W_p(F_\sigma, F) \to 0$ as $\sigma \to \infty$ (Villani, 2009, Thm. 6.9). Similarly we have $W_p(\hat{F}, \hat{F}_\sigma) \to 0$, yielding

$$\limsup_{\sigma \to \infty} D_{0,\sigma} \le W_p^p(F, \hat{F}). \tag{45}$$

Applying the triangle inequality in the reverse direction gives

$$W_p(F, \hat{F}) \le W_p(F, F_\sigma) + W_p(F_\sigma, \hat{F}_\sigma) + W_p(\hat{F}_\sigma, \hat{F}). \tag{46}$$

Taking limits yields

$$\liminf_{\sigma \to \infty} D_{0,\sigma} \ge W_p^p(F, \hat{F}) \tag{47}$$

and the theorem. $\qquad\square$

It follows from the previous result that when the source ensemble is ergodic, as occurs with textures, then in the large-$\sigma$ limit Wasserstein distortion reduces to the ensemble form of realism. That is, it equals the Wasserstein distance (to the $p$th power) between the true distributions of the images and reconstructions, denoted by $F$ and $\hat{F}$ in the following corollary.

**Corollary C.4.** *Suppose $q$ satisfies **P.1**, **P.2**, and **P.6** and $d$ is a metric. Suppose $\mathbf{X}$ and $\hat{\mathbf{X}}$ are stationary ergodic processes and let $F$ (resp. $\hat{F}$) denote the CDF of $Z_0$ (resp. $\hat{Z}_0$). If*

$$\mathbb{E}\left[d^p(Z_0, 0)\right] < \infty \text{ and } \mathbb{E}\left[d^p(\hat{Z}_0, 0)\right] < \infty, \tag{48}$$

*then we have*

$$\lim_{\sigma \to \infty} D_{0,\sigma} = W_p^p(F, \hat{F}) \quad a.s. \tag{49}$$

*Proof.* Among the hypotheses of Theorem C.2, (26) and (27) hold a.s. by the ergodic theorem (e.g., Durrett (1996, Thm 6.2.1)) and (48), and (28) holds a.s. because

$$\mathbb{E}\left[\sum_{k=-\infty}^{\infty} q_\sigma(k) d^p(Z_k, 0)\right] = \sum_{k=-\infty}^{\infty} q_\sigma(k) \mathbb{E}\left[d^p(Z_k, 0)\right] = \mathbb{E}\left[d^p(Z_0, 0)\right] < \infty, \tag{50}$$

by monotone convergence and (48), and similarly for $\hat{Z}$. $\qquad\square$

