# OpenReview forum: "Wasserstein Distortion: Unifying fidelity and realism"
_ICLR.cc/2024/Conference — Submitted to ICLR 2024_

### Official Review · Reviewer_xJ9L · 2023-10-29

**Soundness:** 2 fair
**Presentation:** 3 good
**Contribution:** 2 fair
**Rating:** 3
**Confidence:** 3

**Summary:**

The paper introduces a notion of Wasserstein distortion that unifies fidelity and realism, by varying its input parameters.

**Strengths:**

The approach is well-motivated and the Wasserstein distortion appears to be a proper treatment.

**Weaknesses:**

I feel that the techniques in this manuscript are more in an engineering way. That is to say, in the theoretical aspect, the paper is not that strong. It presumes a lot of domain knowledge in image processing, at least in its current exposition. It might be more suitable for CVPR.

**Questions:**

See my comments above. I feel that the approach needs more stronger theoretical justifications.

---

> ### Author Response · Authors · 2023-11-17
>
> Thank you for the review. We are not aware of any background knowledge from image processing that is assumed in the paper. Could you mention some specific examples? Will we respond to your other comments later.

---

> ### Author Response · Authors · 2023-11-19
>
> Thank you for your review. We agree that the contributions of the paper are not purely theoretical. We do not see this as a weakness, however. We see Wasserstein distortion as a generalization/synthesis of several lines of prior work on models of the human visual system, probabilistic texture models, and image realism/distortion.
>
> The careful insights provided by the reviewers prompted us to replace the existing theorems (which are moved to the appendix) with a result and discussion on the metric properties of Wasserstein distortion and their connection to the spectral properties of the pooling PMF. In particular, we show that Wasserstein distortion is not a valid metric under popular choices of the pooling PMF due to the presence of spectral nulls. We feel that this provides a stronger theoretical justification for the approach and results in a stronger paper overall.
>
> We believe our paper is self-contained, and could not identify any image processing concepts or terms that are relied upon in the paper that require domain knowledge.

---

### Official Review · Reviewer_nULJ · 2023-11-01

**Soundness:** 3 good
**Presentation:** 3 good
**Contribution:** 2 fair
**Rating:** 5
**Confidence:** 3

**Summary:**

The paper introduces a distortion measure between a reference image and a reconstructed image. The motivation of the proposed Wasserstein distortion distance measure is to simultaneously address pixel-level fidelity as well as realism based on the theory of human vision system.

From the reference image, a sequence of probability measures is defined. Each measure in the sequence represents the statistics of the features pooled across a region centered at a location. The given reference image is covered with overlapping pooling regions of various sizes. Each region is associated a distribution obtained by computing various features at random locations within the region. The overall distortion between two images is defined to be the Wasserstein distance between the distributions summed over the pooling regions. The proposed distortion measure reduces to pure fidelity and pure realism as the size of the pooling region tends to zero and infinity, respectively. This enables generation of textures that have high fidelity to a reference texture in one location of the image and a smooth transition to an independent realization of the texture as one moves away from this point.

**Strengths:**

The idea of using nonuniform weights over pixels in distributions enables incorporating fidelity and realism into a common framework

The proposed scheme enables smooth interpolation between fidelity and realism.

The proposed methodology is grounded in theories of the HVS.

**Weaknesses:**

The main contribution of this work over prior works is that the proposed formulation considers distributions using nonuniform weights over pixels while existing works use equal weights. It is not clear to me if the contribution is significant enough. Can one apply simpler modifications to existing approaches to achieve fidelity in specific regions and realism in the other regions? For instance, one could adapt the approach of [Ref1] which can control the spatial distribution of textures according to user-specified annotation maps. I do agree that the proposed Wasserstein distortion enables smooth interpolation between fidelity and realism however, it needs to be demonstrated that the proposed approach is superior to other possible methods.

Comparison in terms of computational requirements of the proposed scheme and existing approaches could have been discussed.

[Ref1] Neural Texture Synthesis With Guided Correspondence CVPR 2023

**Questions:**

While I am not an expert in this topic, I would like to know if the authors could have compared with other methods on standard test sets.

---

> ### Author Response · Authors · 2023-11-19
>
> Thank you for your thorough and incisive review. In response to the weaknesses that you highlighted, we updated our paper to now speak to the utility of nonuniform weights in several places:
>
> We modified Section 3, and in the revised version we show that uniform weights do not result in a valid metric, because there are examples in which D(x,x’) = 0 even though x and x’ are different. Thus one cannot capture pure fidelity with this choice (note that most papers that use uniform weights are interested in realism, not fidelity). We show that the problem lies with the spectral properties of the uniform PMF. We added a new theorem, Theorem 3.1, in which we proved that this problem can be solved if one moves to nonuniform PMFs.
>
> In experiment 1 in the current version (which was in Appendix B.2 in original submission), we find that nonuniform PMFs allow for a smooth transition from fidelity to realism, with an extensive intermediate regime. When using uniform PMFs, the transition is much more abrupt, to the extent that one does not obtain images that balance the two competing goals.
>
>
> It is also worth noting that one cannot smoothly interpolate between fidelity and realism using uniform weights due to integer effects, which are especially pronounced in the regime of small pooling regions (such as when one moves from a region consisting of a single pixel to a region consisting of a 3x3 patch). Finally, we do not see the use of nonuniform weights as necessarily the primary contribution of the paper. Rather, we see our contribution as the identification of a distortion metric, in the mathematical sense, that subsumes fidelity and realism and allows for interpolation between them.
>
>
> In Figure 8 in the original submission (Figure 13 in current version), we illustrated that a simple linear combination of pure fidelity and realism does not achieve the result we have; other simpler modifications, including [Ref1], would require more hand-tuning than we require: [Ref1] requires segmentation, and particular choices of guidance maps for fidelity and realism regions, and an indicator for each patch to be either fidelity or realism. Achieving smooth transitions would need yet more guidance maps. Our method only requires identifying the high fidelity region, which is either automatically calculated from the saliency map provided with the image, or automatically calculated once we know where the center of vision is. Also, we achieve a smoother transition than segmenting the image and processing each patch individually, as we show in the Figures 3,4,6,7 in the original paper (Figures 5,6,8,9 in the revised version).
> In comparison to other papers, multiple reference images are taken from standard references, Gatys et al. 2015, Ustyuzhaninov et al. 2017, etc. Comparing the results in these papers, we achieve comparable performance, even though our aim is not texture synthesis per se.
>
>
> Some comparison of the computational requirements was mentioned in the discussion of Experiment 1 (now moved to Section B.1).

---

> > ### Comment · Reviewer_nULJ · 2023-11-22
> >
> > Thank you for your  response. I feel my concerns regarding experimental comparisons still hold. Therefore I stick with my original rating.

---

> > > ### Author Response · Authors · 2023-11-22
> > >
> > > Thank you for your feedback. We have revised our submission to include a comparison (Appendix B.5, p. 18) of our method to the Sliced Wasserstein distance in [Ref2]. We hope this would address your concern and illustrate our utility.
> > >
> > > [Ref2] Eric Heitz, Kenneth Vanhoey, Thomas Chambon, and Laurent Belcour. A sliced wasserstein loss for neural texture synthesis. In *Proceedings of the IEEE/CVF Conference on Computer Vision and Pattern Recognition*, pp. 9412-9420, 2021

---

### Official Review · Reviewer_fzyQ · 2023-11-02

**Soundness:** 3 good
**Presentation:** 3 good
**Contribution:** 1 poor
**Rating:** 3
**Confidence:** 5

**Summary:**

The authors propose optimal transport as a way to unify pixel fidelity and realism understood as perceptual quality. The unifying nature of the work is the introduction of the construction of a loss function depending on a pooling parameter which in its limits toward 0 and $+\infty$ recover a loss that corresponds respectively to fidelity and realism. These results are backed up by theorems. Finally the authors conduct different numerical experiments : texture synthesis, foveated texture synthesis and reproduction of natural image with saliency maps.

**Strengths:**

- the paper is well grounded in its field, the introduction covers broadly the existing literature
- the structure of the paper is clear and easy to follow
- theoretical claims are illustrated by numerical experiments

**Weaknesses:**

- the Wasserstein distortion as introduced in section 2 is not really new and is in fact very close to the work of Freeman et al (2012) the main innovation being that the parameter sigma can freely be fixed at any position in the image instead of being constrained by the eccentricity of the visual receptive fields.
- the maths of section 2 are overly complicated for a naive reader : in the end the authors use discrete optimal transport between empirical distributions and assume Gaussiannity which in summary corresponds to adjust the mean and standard dev of the local features to a new image (initialized from a white noise image) to the mean and standard dev of the local features of an exemplar image.
- the pooling distribution $q_\sigma$ corresponds to a local weighting of the statistics with width $\sigma$ (as in Freeman 2012).
- when section 2 is well understood the theoretical results become trivial : (i) in the large $\sigma$ limit this is standard texture synthesis framework (Portilla-Simoncelli, Gatys). The specific setting of equation (12) has been empirically evaluated for texture synthesis against Gatys and Portilla-Simoncelli by Vacher et al (Neurips 2020). (ii) in the small $\sigma$ limit this is the exact reconstruction of an image from its feature
- the numerical experiments illustrate the nature of sigma but I do not see in which it contributes to vision study (even in terms of methods)

**Questions:**

No specific questions.

details :
-equation (2) is not a convolution, it becomes convo in eq (3)
-the title is a bit misleading by talking about fidelity and realism, it is a bit disappointing that this corresponds to notion of pixel distance vs perceptual distance...

**Post-response update**
Soundness have been increased from 2 to 3.
I would have increased my score to 4 but I can't and I think there is to much to be done to really be just under the acceptation threshold.

---

> ### Author Response · Authors · 2023-11-17
>
> We are not sure what paper "Freeman et al. (2012)" refers to. Is the reviewer referring to
> J. Freeman and E. P. Simoncelli, “Metamers of the Ventral Stream,” *Natural Neuroscience*, vol. 14, pp. 1195-1201, 2011?

---

> ### Author Response · Authors · 2023-11-19
>
> Thank you for your review. We would respond to the weaknesses you mention as follows:
> We assume the reference is to the paper:
> J. Freeman and E. P. Simoncelli, “Metamers of the Ventral Stream,” Natural Neuroscience, vol. 14, pp. 1195-1201, 2011.
>
>
> This paper is indeed the point of departure for our work, as noted in the introduction. Our paper extends Freeman and Simoncelli in several ways:
>
>     Freeman and Simoncelli do not provide a distortion measure or loss function. They create perceptually indistinguishable images ("metamers") by constraining them to have the same local statistics across different patches. Thus the optimization problem they are solving has no objective; their goal is to find a feasible point, i.e., one that satisfies their constraints. Under their approach, whether two images are metamers is a binary notion; they do not quantify how far two images are from being metamers. Beyond being able to measure how close images are to being metamers (as in current Figure 12, Figure 10 in original submission), defining an explicit distortion measure enables the idea of measuring discrepancy between local statistics to be applied more broadly, such as in compression applications. It also allows one to precisely answer structural questions about the distortion measure, such as what properties of the pooling PMF we require, which brings us to the next point.
>
>     Section 3 of the paper has been replaced with a discussion of what properties we require in order for Wasserstein distortion to be a proper metric (in particular, to satisfy that d(x,x’) = 0 iff x = x’). We find that both the raised-cosine-type PMF used by Freeman and Simoncelli and the uniform PMF used in most other works is problematic in this regard. We explain why and suggest PMFs that are superior for our purposes.
>
>     Freeman and Simoncelli assume that the viewer focuses on the center of the image, and the pooling regions are the same for all images. We provide an automatic method for producing an image-specific sigma map from a given saliency map.
>
>     We connect the pooling idea of Freeman and Simoncelli to the recent literature on the distortion-perception tradeoff. In that literature, distortion and perception (or realism) are viewed as disparate constraints that are fundamentally different in character. The main goal of this paper is to show that they can be lifted in a common framework, inspired by Freeman and Simoncelli. The paper thus has the potential to change the way both the machine learning and information theory communities view the distortion-perception tradeoff.
>
>
> We believe that we made the math in Section 2 as simple as possible and that it is inline with other ICLR papers. The introduction provides a non-mathematical description of the ideas as a warm-up, and the paper is not especially notation-heavy by the standards of the community. We define Wasserstein distortion and prove our theorems in 1-D for simplicity’s sake. Note that we do not make a global Gaussian assumption. In particular, Theorems 3.1 and 3.2 in the original submission (which are moved to Appendix C now) make no reference to the Gaussian distribution. Our experiments resort to means and variances for computational reasons, but one cannot pose Wasserstein distortion in terms of means and variances without compromising the generality of the theorems.
>
>
> The sigma parameter does indeed correspond to the width of the pooling region in Freeman and Simoncelli, but we do not view this as a weakness of the paper.
>
>
> While we would not describe Theorems 3.1 and 3.2 in the original submission as trivial (they require proofs, and the proofs are more than a few lines), they are intended to confirm that Wasserstein distortion has the extremal properties that one intuitively expects. We have moved these theorems to the appendix to avoid giving the impression that these results are difficult or unexpected. We replaced them with a result and discussion on how the metric properties of Wasserstein distortion relates to the spectral properties of the pooling PMF, which we believe will be of greater interest.
>
>
> We are not sure what “vision study” refers to. The goal of the experiments was to validate the proposed distortion measure using the method propounded by Ding et al. (2021), namely by generating random images that are close to a given image under the proposed measure. In particular, we sought to illustrate how it can be useful to interpolate between fidelity and realism constraints. The goal was not to contribute to the literature on the human visual system. We have replaced Experiment 1 with a different experiment that interpolates between fidelity and realism without any reference to peripheral vision, which hopefully helps clarify this point.

---

> > ### Comment · Reviewer_fzyQ · 2023-11-19
> > **Response**
> >
> > Thanks for your response and for clarifying few things that I might have missed.
> >
> > I was indeed referring to Freeman *et al.* (2011). I apologize for the mistake in the year.
> >
> > **Objectives of the work**
> > I am a bit confused by the nature of the paper. It seems like it is a method/theory paper for vision studies (and I am referring broadly to research interested in human/primate vision from cognition to neuroscience) but at the same time the results are embedded in optimal transport theory which will make it cryptic for people in this field. In addition, in your response your are explaining how your inspiration comes from Freeman *et al.* and that it is important to extend their work. Freeman *et al.* created their model for studying metamers and they are conducting experiments. In the end, you are telling me that the *goal was not to contribute to the literature on the human visual system*. Well, then why are you working on this ? Why is it important to quantify metamerism ? Why it should be a mathematical distance ? The paper in its current version does not demonstrate that.
> >
> > **About loss function**
> > This is not because there is no objective loss function written in Freeman *et al.* (as in Portilla & Simoncelli 2000) that there is no objective. There is a clear objective and it is the L2 norm between the vector of summary statistics (pooled over some areas in Freeman *et al.*). Back in 2000, there were no widespread auto-diff so the design of the optimization relies on smart engineering. Yet, you can now find a pytroch implementation based on the definition of a single objective function (https://github.com/LabForComputationalVision/plenoptic/blob/main/examples/Metamer-Portilla-Simoncelli.ipynb). So, to me, your contribution is only to make Freeman *et al.* approach more flexible by defining a local pooling width no constraint by eccentricity.
> >
> > > Freeman and Simoncelli assume that the viewer focuses on the center of the image, and the pooling regions are the same for all images. We provide an automatic method for producing an image-specific sigma map from a given saliency map.
> >
> > I agree that the fixed pooling regions is a limitation. Specifically I have two references in mind :
> > 1. Wallis, T. S., Funke, C. M., Ecker, A. S., Gatys, L. A., Wichmann, F. A., & Bethge, M. (2019). Image content is more important than Bouma’s Law for scene metamers. ELife, 8, e42512.
> > 2. Vacher, J., Launay, C., & Coen-Cagli, R. (2022). Flexibly regularized mixture models and application to image segmentation. Neural Networks, 149, 107-123.
> >
> > The first reference show that in the visual periphery there are things that are more robust and cannot be considered as pooled in a larger region like contours between visual object for example. The second reference is a probabilistic model of visual segmentation which attempt to perform statistical grouping over features used for texture synthesis. So basically, it looks for unfixed pooling regions build upon statistical similarity. To me, your work might contribute to this line of research.
> >
> > **About trivial results**
> > The mathematics are written with clarity and precision and the proofs are correct. It demonstrates that the authors have clear understanding of what they are doing. Though, the level of formalism can be a problem depending on the targeted readership (and here it's unclear to me...). In addition, the length of a proof is not a criteria for triviality. I am considering the result trivial because in the limit of small sigmas your problem is the transport between a single Dirac distribution (so of course the features will exactly match at every single pixel position) while in the large sigmas this is the regular discrete optimal transport setting (so only the empirical distribution will be the same). That is ok to state it but the result is not so deep because transport between weighted sum of Dirac is a standard setting (see Chapter 2 of Peyré & Cuturi 2019).
> >
> > Peyré, G., & Cuturi, M. (2019). Computational optimal transport: With applications to data science. Foundations and Trends® in Machine Learning, 11(5-6), 355-607.
> >
> > **Conclusion**
> > I think the authors have to make a choice about the motivation underlying their work. If it's not a contribution to visual perception studies then I consider that the contributions are very little. If it is then I see where the work could go but I would expect to see some perceptual data or at least a proof of concept to demonstrate how it can solve some problem in the field. I have update my scores accordingly and I encourage the authors to pursue their efforts if the paper were not accepted.

---

> > > ### Author Response · Authors · 2023-11-20
> > >
> > > We appreciate the prompt and detailed response. We understand the nature of the reviewer’s concerns now and are better able to address them.
> > >
> > > **Regarding the objectives:** the title, abstract, and introduction place the work squarely in the recent literature on the distortion-perception tradeoff. This includes papers such as:
> > > - Yochai Blau and Tomer Michaeli. The perception-distortion tradeoff. In Proc. IEEE Conference on Computer Vision and Pattern Recognition, pp. 6228–6237, 2018.
> > > - Yochai Blau and Tomer Michaeli. Rethinking lossy compression: The rate-distortion-perception tradeoff. In Proceedings of the 36th International Conference on Machine Learning (ICML), pp. 675–685, 2019.
> > > - Lucas Theis and Aaron B. Wagner. A coding theorem for the rate-distortion-perception function. In Neural Compression: From Information Theory to Applications – Workshop @ ICLR 2021, 2021. URL https://openreview.net/forum?id=BzUaLGtKecs
> > > - Aaron B. Wagner. The rate-distortion-perception tradeoff: The role of common randomness. arXiv preprint arXiv:2202.04147, 2022. doi: 10.48550/arXiv.2202.04147.
> > > - Jun Chen, Lei Yu, Jia Wang, Wuxian Shi, Yiqun Ge, and Wen Tong. On the rate-distortion-perception function. IEEE Journal on Selected Areas in Information Theory, 2022. doi: 10.1109/JSAIT.2022.3231820.
> > > - George Zhang, Jingjing Qian, Jun Chen, and Ashish Khisti. Universal rate-distortion-perception representations for lossy compression. Advances in Neural Information Processing Systems, pp. 11517--11529, 2021.
> > >
> > > The community interested in this topic lies in the intersection of image processing (especially image compression), information theory, and machine learning, and it is mathematically-oriented. Within the image processing and information theory communities in particular, the need for good distortion metrics has been recognized for so long as to be unquestioned, which is why we do not explicitly motivate it in the paper. The above papers trace their lineage to historical papers focused on distortion measures such as
> > > - Zhou Wang, Alan Conrad Bovik, Hamid Rahim Sheikh, and Eero P. Simoncelli, Image quality assessment: From error visibility to structural similarity, IEEE Trans. Image Proc., Vol. 13, No. 4, pp. 600-612, 2004.
> > >
> > > in image processing and
> > > - James L. Mannos and David J. Sakrison, The effects of a visual fidelity criterion on the encoding of images, IEEE Trans. Inform. Theory, Vol. 4, pp. 525–536, 1974.
> > >
> > > in information theory. The former is one of the most important papers in its field, with over 47K citations according to Google Scholar. The paper was informed by contemporary HVS studies, yet its contribution lies in proposing a distortion measure for engineering purposes. The measure is now used in a myriad of engineering applications and has won industry awards.
> > >
> > > Likewise, the goal in our paper is to draw from, rather than contribute to, the HVS literature for purposes of developing a metric.  Contributions we make to the HVS literature would be serendipitous and incidental. We view our paper as belonging to the above literature on distortion-perception, and we hope it can be reviewed as such. We do not see it as a method/theory paper for vision studies as the reviewer suggests.
> > >
> > > **Regarding the loss function:** While one can debate how explicit the loss function is in Freeman and Simoncelli, their paper does not address the spectral issues of the weighting function now described in Section 3. In fact, their proposed weighting function has poor spectral properties, as noted therein.
> > >
> > > **Regarding the triviality of the results:** We appreciate the reviewer’s detailed response, and we understand the source of the reviewer’s concern better now. We believe it comes about from a misreading of the theorem (Theorem 3.1 in the original submission; Theorem C.1 in the revision). The assertion is not that Wasserstein distortion reduces to a pure fidelity measure when one sets sigma = 0. That is an entirely trivial statement that is noted in passing immediately before the theorem (Eq. (18) in the revision), to contrast it with the theorem. The assertion in the theorem itself is that Wasserstein distortion continuously reduces to a pure fidelity measure in the limit as sigma -> 0. Mathematically, this is a continuity statement that is less trivial. As articulated in the paper, since we are interested in smoothly interpolating between the two regimes, it is the continuity result that is of interest to us. Likewise for the sigma -> infinity regime.
> > >
> > > We thank the reviewer again, and we hope they are willing to reassess the paper with the above in mind.

---

### Official Review · Reviewer_GTX1 · 2023-11-06

**Soundness:** 3 good
**Presentation:** 3 good
**Contribution:** 3 good
**Rating:** 5
**Confidence:** 4

**Summary:**

The authors propose a measure of image distortion based on an optimal transport approach, under the dual constraints of fidelity and realism. Fidelity is expressed by a distance metric such as PSNR, and realism by comparison to a collection of patches. The authors also propose to use a model of the Human Visual System (HVS) with a foveal-like receptive field to strike a varying balance between fidelity and realism depending on a distance to a given point or to a saliency map. They use their distortion measure for texture and realistic image synthesis.

**Strengths:**

The paper is well written and clear. The objective of proposing a distortion measure associated with characteristics of the HVS is interesting and novel. The formulation of their distortion measure based on Wasserstein distances is sound and also practical. The authors achieve good and efficient texture synthesis results but not as impressive synthesis of realistic images as recent methods based on diffusion. Nonetheless their approach is much more interpretable. They do use features extracted from a VGG-19 network for visual realism measurement, but this network has been well studied and could be understood as interpretable as well.

The authors have provided proofs and code in appendix, and so their results should be reproducible.

**Weaknesses:**

The paper is mostly a proof-of-concept at this stage. The author mention using their method for image encoding but this is not part of this work, and it is not clear that it would work well since, like recent super-resolution approaches based on single image patches approach, it could tend to fill in details with elements that would look realistic but are not real.

**Questions:**

- The foveal approach is interesting but tends to produce quickly degrading results far from the high saliency results in figure 4 and similar. The HVS is not reducible to a degradation of resolution far from the fovea, it accepts less than perfect results but consistency is also important. Could the approach be improved by iterating over the generated image to identify highly unrealistic portions (such as broken lines, etc) and refine those? How could this made to work?

- Could the distortion measure be used to identified generated or edited content in an image, say to identify tampered images?
- How fast doe the proposed algorithm work, say to produce a 512x512 image?

---

> ### Author Response · Authors · 2023-11-19
>
> Thank you for your careful and in-depth review. We especially appreciate your summary and description of the strengths of the paper. Regarding the weakness you mention, hallucination would appear to be intrinsic to any scheme that seeks a high level of realism without enforcing high pixel-level fidelity. Yet, image generation by minimization of Wasserstein distortion is less prone to hallucinating realistic elements that bear no resemblance to the original image because it relies entirely on local statistics of the image, without reference to the ensemble of all images. If the original image is, say, blurry, then we expect the reconstructed image to be blurry as well. In this sense the weakness you mention is less applicable to Wasserstein distortion than other methods of achieving realism, such as diffusion models. This was discussed in the last paragraph of the original submission. Also note that Wasserstein distortion, as a distortion measure derived from image statistics, is designed to quantify the extent to which a reproduced image contains realistic elements that are disconnected from the reference image. Rather than producing images that withstand close scrutiny in all regions, our image generation experiments are meant to demonstrate the properties of our metric, namely that it becomes increasingly permissive to error in the periphery, and that these errors are hard to spot when viewing the image at an appropriate distance.
>
>
> We would answer your questions as follows:
>
> In our experiments 2 and 3, the rate at which sigma grows as one moves away from the fovea is a parameter that we control. If we tune the parameter such that sigma grows slower, the reproduction would degenerate less. Thus the approach does not necessarily need to produce quickly degrading results. We do, in fact, iterate over the images in that we measure the Wasserstein distortion at a subset of points, for computational reasons, and this subset is randomly selected from iteration to iteration. The lack of consistency with respect to lines seems to be an artifact of our use of VGG19. Other texture generators that use VGG19 have similar limitations, and various papers have proposed ways of augmenting VGG19 to address this (see the references in the caption for Figure 7 in the original submission which is Figure 9 in the revision). This theory is supported by the final row of Figure 9, which is new in the revision. This is simply a repeat of the third row but with a downsampled image. We see that the foul lines are much more consistent, presumably because VGG-19 is better able to capture long-range dependence on the downsampled image. We emphasize that our method is agnostic to the choice of features and the various improvements to VGG-19 cited in the caption to Figure 9 could in principle be incorporated into our framework.
>
> Wasserstein distortion cannot detect generated or edited content because it is a full-reference distortion measure. Thus it can only measure the distance between two images. It cannot score the likelihood or realism of a single image in isolation.
>
> For the most complicated experiment we have, namely experiment 3, it took about 3 hours per image (which are 480x480) on an NVidia GTX4090 to go from a completely random image to the final image shown in the paper. We added this information in Appendix B in our revised submission. Some discussion of complexity is included in Appendix B.1 (which was included in the paper itself in the original submission).

---

### Author Response · Authors · 2023-11-17
**Initial response**

We wish to thank all four reviewers for their constructive comments on the paper. We agree with the view expressed in several of the reviews that the original submission did not include an adequate discussion of why this paper represents an advance over prior work, especially Freeman and Simoncelli (2011). We would like to summarize here why we believe it does indeed represent an advance, and in particular why the choice of the weighting PMF, which a core novelty of the paper, is important and nonobvious.

Freeman and Simoncelli do not provide a distortion measure or loss function.  They create perceptually indistinguishable images ("metamers") by constraining them to have the same local statistics across different patches. Thus the optimization problem they are solving has no objective; their goal is to find a feasible point, i.e., one that satisfies the constraints.  Under their approach, whether two images are metameters is a binary notion; they do not quantify how far two images are from being metamers.

One might argue that the distortion measure obtained by summing the divergences between local statistics over patches is suggested by Freeman and Simoncelli although it is not explicit in their paper.  Yet much hinges on the choice of the weighting PMF within the patch, and this is not recognized by Freeman and Simoncelli or other papers that use similar ideas.  Consider, for simplicity, a uniform weighting over a patch.  This is essentially Wasserstein distortion with a uniform PMF, such as [1/3, 1/3, 1/3] if one were to consider a patch of size 3 in 1-D. This does not work for several reasons:

1) It does not result in a valid metric, even in feature space, in that there are adversarial examples of pairs of distinct images whose distortion is exactly zero. This arises due to spectral nulls of the PMF that occur when the PMF is uniform. By allowing nonuniform PMFs, we can ensure that the PMF has no spectral nulls. This ensures that the distortion is nonzero for images that differ, by any amount, in the feature space.

2) In our experiments, we find that if one relies on uniform PMFs, then as one varies the patch size the distortion measure toggles abruptly between acting like a pure fidelity constraint to acting like a pure realism constraint, with little intermediate behavior.  In contrast, by using nonuniform PMFs with a graceful falloff, one can achieves a smooth transition, with an interesting intermediate regime in which both goals are met to some extent (cf. Fig. 9 in the original submission).

3) Mathematically, relying on uniform PMFs over patches of different sizes cannot continuously interpolate between fidelity and realism because patch sizes must be integer-valued.

Freeman and Simoncelli do consider nonuniform PMFs (cf. their equation (9)), but they do not confront any of the above issues. In particular, their recommended PMF has poor spectral properties.

We are in the process of revising the paper to reflect the above.  We intend to upload tomorrow. We believe that the revision will be significantly stronger than the original version, thanks to the critiques offered by the reviewers. We will comment further when the new version is uploaded. Thank you again for your comments.

---

### Author Response · Authors · 2023-11-19
**Revised Paper Uploaded**

We have uploaded a revised paper that includes the changes described in the previous comment, among others. We hope the reviewers are able to take a fresh look at the paper, as we believe that the contributions are now much clearer, thanks to the critiques offered by the reviewers.

---

### Meta-Review · Area_Chair_qWXr · 2023-12-06

**Metareview:**

The reviewers acknowledged the strengths of the paper, noting the proposal of novel approach to incorporating the characteristics of the human visual system into image synthesis. They however raised several concerns that preclude acceptance. They all agree that, despite the addition of new results in the rebuttal, this works lacks some more in-depth numerical evaluation. The application of the method to image encoding, though mentioned, was not demonstrated in the paper, and this would be a welcome addition. Also, the high level of theoretical complexity was also noted as concerns by some reviewer, so that the writing style could be improve to make the paper accessible to a wider audience in the field of  visual perception studies.

**Justification For Why Not Higher Score:**

I think this paper is quite far bellow the acceptance bar.

**Justification For Why Not Lower Score:**

N/A

---

### Decision · Program_Chairs · 2024-01-16

Reject